# Behavioral shifts mask the success of legislation and outreach for endangered species recovery

Victoria J. Bakker [1,11] ✉, Daniel F. Doak[2], Alacia Welch[3], L. Joseph Burnett[4], María C. Porras Peña [5], Joseph Brandt[6], Sharon A. Poessel [7], Steve Kirkland[6], Rachel Wolstenholme[3], Daniel Ryan[3], Mike Stake[4], Arianna Punzalan[6], Nacho Vilchis[8], Melissa A. Braham[9] & Myra E. Finkelstein [10,11] ✉

A fundamental challenge in conservation is assessing the efficacy of recovery actions to optimize endangered species management. Considerable recent attention has focused on effective measures to counter the endangerment of avian scavengers, which have declined worldwide, primarily due to poisoning. One iconic example is efforts to recover the critically endangered California condor (*Gymnogyps californianus*), whose leading cause of death is poisoning from ingesting lead-based ammunition in carcasses. Despite enormous resources expended in California, USA, including implementation of public outreach campaigns and two legislative bans on lead ammunition, lead-related mortality of condors has increased. Here we show that two types of behavioral shifts explain the observed increases in condor lead exposure: wilder foraging and ranging by condors and increased shooting of wild pigs (*Sus scrofa*) by humans. After accounting for these trends, we show that both lead ammunition bans and public outreach efforts have significantly reduced condor blood lead levels in California, lowering mortality. Our analyses uncover a dynamic in which changing ecological conditions mask the true efficacy of legislation and outreach. Given rapid global change, such dynamics are likely operating in many settings, underscoring the importance of comprehensive evaluations of recovery actions, which can be obscured by shifting behaviors and threats.

A major challenge to preserving our remaining wild species is ensuring that endangered species recovery actions yield maximum results[1]. Quantifying the efficacy of recovery actions is challenging, especially for long-lived, slow-reproducing species, because decades may be required to detect improved population trends[2]. Although some actions can produce rapid and easily quantifiable effects (e.g., captive-breeding and release[3–5], invasive species removal[6–9], others operate more slowly or are less clear to evaluate (e.g., public outreach[10–12], habitat restoration[13,14]). Recovery actions also take place in nonstationary ecological settings, in which behavioral shifts by recovering or co-occurring species, as well as by humans, can increase threat levels, masking the success of even highly effective measures. Given the

[1]Department of Ecology, Montana State University, Bozeman, MT, USA. [2]Department of Environmental Studies, University of Colorado, Boulder, CO, USA. [3]National Park Service, Pinnacles National Park, Paicines, CA, USA. [4]Ventana Wildlife Society, Monterey, CA, USA. [5]Parque Nacional Sierra de San Pedro Mártir, Ensenada, Baja California, Mexico. [6]U.S. Fish and Wildlife Service, Ventura, CA, USA. [7]U.S. Geological Survey, Forest and Rangeland Ecosystem Science Center, Boise, ID, USA. [8]Conservation Science & Wildlife Health, San Diego Zoo Wildlife Alliance, San Diego, CA, USA. [9]Conservation Science Global Inc., Cape May, NJ, USA. [10]Department of Microbiology and Environmental Toxicology, University of California, Santa Cruz, CA, USA. [11]These authors contributed equally: Victoria J. Bakker, Myra E. Finkelstein. ✉e-mail: vjbakker@gmail.com; myraf@ucsc.edu

speed of global change, such dynamics are likely to play an increasingly prominent role in species conservation, and correctly understanding these opposing effects will be particularly important when conservation actions are costly or controversial.

For reintroduced populations, understanding how changing behavior influences population dynamics can be critical[15–17]. An often-implicit assumption of endangered species recovery is that restoring wild behavioral traits (i.e., natural behaviors typical of wild-born free-ranging conspecifics) for released individuals will increase their fitness[18–20]. While expression of wild behavior is often desirable for species recovery, it can entail moving out of protected areas or utilizing more dangerous resources[21]. Evaluating interactions between wild behavior and threat exposure is increasingly important as conservation seeks to maximize the value of human-modified landscapes[22,23] and some voices call for global rewilding[24–26].

Understanding changes in human behavior, in particular the effectiveness of public outreach, is also of critical importance[27]. Across the globe, many species recovery plans include an outreach component (e.g., U.S. Endangered Species Act, EU LIFE Programme) with the goal of altering human behavior to reduce threats to wildlife. Documented success of such programs, however, is extremely limited[28,29]. While many studies have measured changes in conservation awareness due to outreach or extrapolated changes in demographic rates for target species from assumed or documented changes in human behavior[30], there are exceedingly few that directly link outreach to changes in biologically meaningful metrics of conservation success (e.g., health, survival, abundance, but see ref. [31]).

The challenges of quantifying the efficacy of recovery for long-lived species in changing ecological settings intersect for many imperiled species, including avian scavengers such as vultures, which have declined worldwide[32], primarily due to poisoning[32–36]. One iconic example is the critically endangered California condor (*Gymnogyps californianus*). Lead poisoning from feeding on carcasses containing lead-based ammunition contributed to the complete extirpation of condors from the wild in the 1980s[37]. Enormous resources have been expended over the past three decades to recover condors in the U.S. state of California, where two of the oldest and largest reintroduced flocks reside (*Central* and *Southern*, Fig. 1, Supplementary Fig. 1a–c), including outreach to encourage nonlead ammunition use and two legislative bans on lead-based ammunition. Despite these actions, condor lead-related mortality has increased[38], calling into question the effectiveness of these costly and controversial efforts.

Here, we show that changing behaviors of both humans and condors interacted with and masked the success of recovery efforts for California condors, which have nonetheless been highly beneficial. To quantify the effectiveness of lead bans and public outreach for condor recovery, we use extremely rich long-term datasets (1996–2023) for *Central* and *Southern* flocks in California, including near-daily behavioral observations (n = ~958,000, Supplementary Fig. 1d–h) for all free-flying condors, necropsies for all retrievable condor carcasses (n = 145), county-specific deer and wild pig hunt tag returns and wild pig cull reports (Supplementary Fig. 2c–h), and quantitative records of outreach activities (Supplementary Fig. 2i). We analyze these data to identify factors predicting condor blood lead levels (n = 3179) and survival (n = 449 condors, 251 mortalities) before and after ban implementations. We also estimate the number of contaminated meals through time and extrapolate the effects of outreach on condor survival. Although no demographic or population abundance data exist for condors prior to the use of lead-based ammunition, we show that the reintroduced Baja California, Mexico flock (*Baja*) has not been significantly impacted by lead and thus serves as a control population demonstrating potential survival and population growth rates in the absence of lead exposure.

## Results and discussion

### Condor lead exposure and mortality has increased

Despite three decades of intensive conservation, the condor's annual population growth rate ($\lambda$) in California is below 1.0 ($\lambda = 0.971$ if excluding contributions from captive releases, Supplementary Tables 2 and 7). Consistent with previous reports[39,40], lead poisoning was the leading cause of condor mortality, representing 62% and 44% of known-cause mortalities for *Central* and *Southern*, respectively, since 1996. The proportion of birds lead exposed (>10 μg/dL), lead poisoned (>35 μg/dL), and highly lead poisoned (>100 μg/dL) in biannual blood lead monitoring has increased over time for both California flocks (Fig. 2a). In tandem, their annual survival probability has declined ($S_{Overall}$, Fig. 2b), including survival of lead hazards ($S_{Pb}$, Fig. 2c), providing evidence that increases in blood lead levels have led to increased mortality. In contrast, *Baja* has had very low blood lead levels (Fig. 2a), experienced essentially no lead-related mortality (Fig. 2c), and has an increasing population ($\lambda = 1.017$ if excluding contributions from captive-bred releases, Supplementary Tables 2 and 8). Feathers, which document lead exposures over longer timeframes than blood[39,41], also indicated minimal lead exposure for *Baja*, with very few birds (<7%) having estimated blood lead levels >35 μg/dL (versus >50% of California-based birds, Fig. 2a).

To better understand the paradox of rising condor lead exposures in California despite significant efforts to limit lead-based ammunition use, we fit linear mixed effects models (LMMs) and calculated variable importance metrics (Supplementary Fig. 3) to dissect the different and sometimes opposing influences (e.g., condor behavior, hunter behavior, post-ban time periods, outreach effort) on condor blood lead levels (Supplementary Tables 3, 5, 9–12). Given the limitations of blood lead for documenting lead exposure histories[39,41], our models explain a substantial amount of variance in blood lead levels, with conditional $R^2$ values[42] for *Central* and *Southern*, respectively, of 0.383 and 0.348 for lead exposure models using our entire time span of data (Supplementary Table 3) and 0.312 and 0.433 for lead exposure models that include measures of outreach, which were restricted to Bioyears (i.e., Sep–Aug) 2012–2021 for *Central* and 2014–2021 for *Southern* (see "Methods": Public outreach levels, Supplementary Table 5). Analyses of these LMM model sets show that while multiple models had low AICc values, explanatory variables in the best models were generally the ones most supported across all models and the same patterns of effects were stable across the well-supported models (Supplementary Fig. 3, Supplementary Tables 9–12).

### Wilder behavior raised lead exposure

Rising lead exposure has coincided with increasing expression of wild behavior by condors. Two key metrics of foraging and ranging behavior became wilder over the 27-year study period—condors fed less often on lead-free carcasses proffered at feeding stations (i.e., proffered feeding rate, *Proffered*) and also spent less time near release sites (excluding proffered feeding), as detected visually or via radio-telemetry (*Presence*) (Supplementary Tables 1, 3, Supplementary Fig. 1d–g, *Proffered* declined by 52% and 85%, *Presence* declined by 42% and 70%, for *Central* and *Southern* respectively). In addition, coastal birds in *Central* feed on beach-cast marine mammals, which rarely contain ammunition[43], but *Central* birds have on average shifted to more inland space use (*Coastal* declined by 43%, Supplementary Fig. 1h). Condors with high *Proffered* and *Presence* had lower blood lead (Supplementary Tables 3 and 5), suggesting condors are less likely to ingest lead if they feed frequently on proffered carcasses or if they remain near release sites, where either hunting is prohibited or people likely have had longer exposure to outreach efforts and possess greater awareness of condors. Similarly, *Coastal* was associated with reduced blood lead (Supplementary Tables 3 and 5), consistent with marine mammal carcasses representing a low-lead food source.

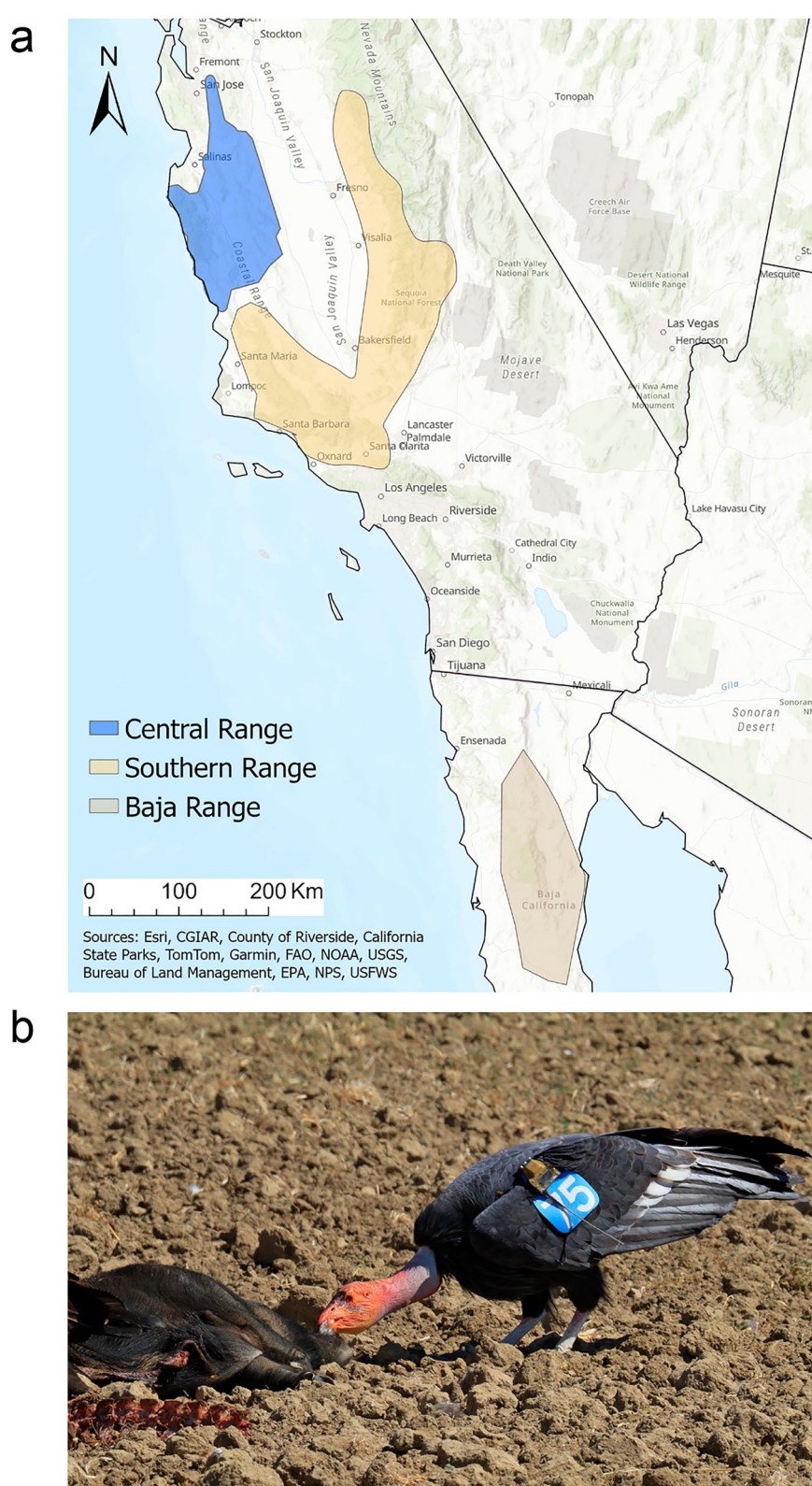

**Fig. 1 | Critically endangered California condor (*Gymnogyps californianus*) populations in the U.S. state of California and the Mexican state of Baja California. a** Condors occur in three flocks in our study area: *Central*, jointly managed by Pinnacles National Park and Ventana Wildlife Society, *Southern*, managed by the U.S. Fish and Wildlife Service, and *Baja*, managed by Comisión Nacional de Áreas Naturales Protegidas and Espacios Naturales y Desarrollo Sustentable. **b** *Central* condor 375 feeding on a wild pig. All free-flying condors are wing-tagged and most have telemetry tags for GPS tracking, with monitoring occurring on a near-daily basis. Photo credit: Gavin Emmons.

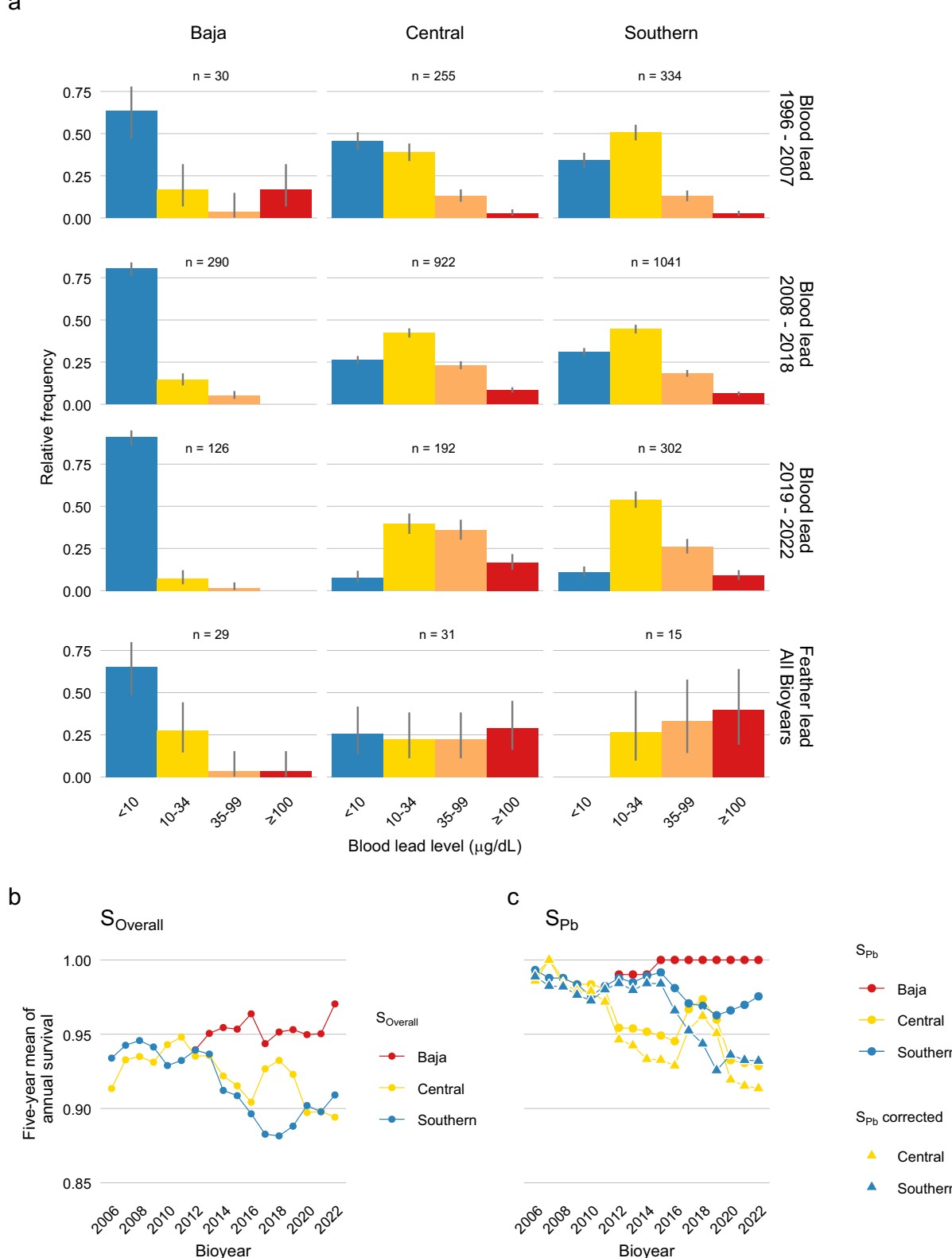

**Fig. 2 | The effects of lead exposure on California condors (*Gymnogyps californianus*).** **a** Frequency of blood lead exposure levels through time for California flocks (*Central, Southern*) vs. Baja California, Mexico flock (*Baja*), based on randomly collected blood and feather samples, with 90% exact binomial confidence intervals. Plotted feather lead is blood lead estimated from peak exposure measured in the feather. Blood lead >35 μg/dL has been associated with adverse health effects in birds and >100 μg/dL is considered potentially lethal[39]. Five-year rolling average of **b** overall survival ($S_{Overall}$) and **c** survival of lead hazards ($S_{Pb}$) by flock, where $S_{Pb}$ (circles) is restricted to deaths directly attributed to lead, and $S_{Pb}$ corrected (triangles) assumes deaths of unknown cause are attributable to lead proportional to deaths of known cause. Source data are provided as a Source data file.

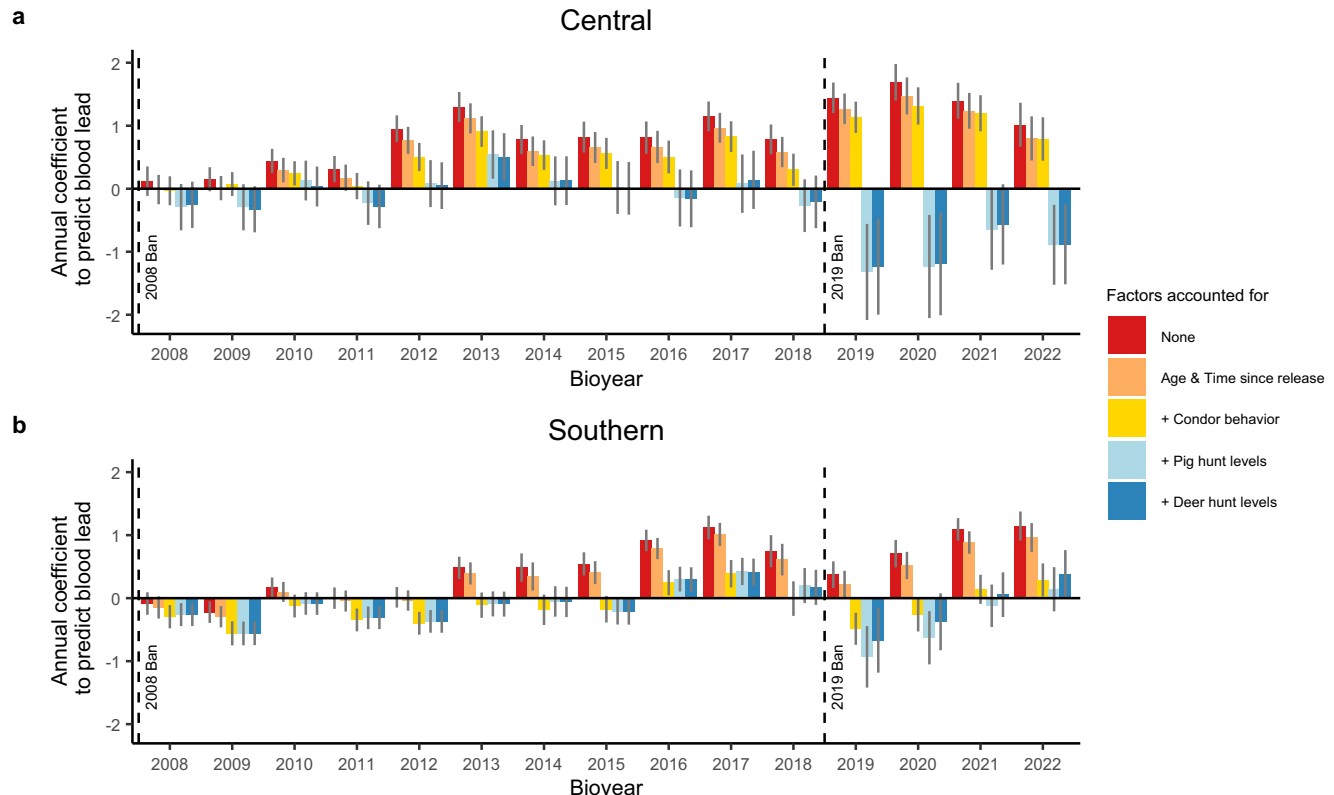

**Fig. 3 | Drivers of blood lead exposure for California condors (*Gymnogyps californianus*) in California.** Plots show categorical year effects (beta coefficients) with 90% confidence intervals for **a** *Central* and **b** *Southern* estimated from successively more complex linear mixed models that predict condor blood lead levels (*lnPb, n* = 1421 samples for 188 *Central* condors, 1758 samples for 179 *Southern* condors) for *Bioyears* (Sep–Aug) 2008 onward, relative to all samples before *Bioyear* 2008. Red bars depict the base model, which accounts only for whether a blood sample was targeted for suspected lead exposure (i.e., *lnPb ~ Int + Targeted + Bioyear + 1 | ID*) and thus shows the observed increasing temporal trend in lead exposure for both flocks. Remaining sets of bars show the residual temporal trends in exposure after accounting for the added effects of other classes of variables (Supplementary Table 3), thus showing the degree to which these variables account for changes over time. Orange bars show year effects for a model that also accounts for age and time-since-release effects. Yellow bars show year effects for a model that adds effects of condor behaviors (e.g., rates of proffered feeding) and

thus illustrate annual differences not accounted for by age, time since release, and the increases in wild behavior of condors. Note that wilder behavior drove most of the lead exposure increases for **b** *Southern*, as indicated by the very weak temporal trends in exposure after accounting for these effects. Finally, blue bars show results for models that add hunting effects and thus show how lead exposure has changed from year to year after accounting for all supported factors linked to lead exposure (i.e., targeted, age and time since release, condor behavior, hunting). Pig hunting (light blue bars, which add *PigHunt* and a seasonal effect) increased lead exposure in both flocks (**a**, **b**) and drove most of the increases for **a** *Central*. Plotting year effects for increasingly complex models unmasks the strong role that the 2008 and 2019 lead-based ammunition bans, depicted by dashed vertical lines, played in lowering exposures for both flocks; after accounting for other factors, annual effects were consistently negative following the bans. Source data are provided as a Source Data file.

Feeding behaviors not captured by *Proffered* and *Presence* may also correlate with additional lead exposure risk. Condors are social feeders and higher dominance at carcasses could increase an individual's likelihood of ingesting a lead fragment. In both flocks, first-year birds, captive-bred birds released <2 years, and birds with fewer recent free-flying days in the wild had lower blood lead, while young adults (6 to 10 years post-fledge) had higher blood lead (Supplementary Tables 3 and 5). Young adults may arrive first or be dominant at carcasses, while newly fledged and newly released birds, and birds that have spent substantial time in captivity, may arrive later or behave subordinately[44,45].

Shifts in these metrics of condor behavior account for essentially all observed increases in blood lead over the last 27 years for *Southern* but not *Central* (Fig. 3), where additional model factors, in particular changes in wild pig hunting, are required to explain increases in blood lead.

**Shifts in shooting behavior masked ban success**

The source of almost all lead exposure for condors is lead-based ammunition[39] found in gut piles from field-dressed carcasses left by

hunters or in intact carcasses left after culling (e.g., wild pigs), pest control (e.g., ground squirrels), or wounding. In California, two legislative bans were passed to restrict the use of lead ammunition—a 2008 ban in condor range for take of big game and nongame (AB 821, Ridley-Tree Condor Preservation Act) and a 2019 ban statewide for take of any wildlife (AB 711, passed in 2013, fully implemented in 2019). Hence, the use of lead ammunition for deer and wild pig hunting as well as control of wild pigs or ground squirrels became illegal in condor range in 2008 and throughout California in 2019. Our analyses detected patterns consistent with shifts in lead-based ammunition use after the 2008 and 2019 bans in both California flocks, including strong and sometimes opposing effects of deer and pig hunt levels, indexed by hunter tag returns, on condor blood lead values (Fig. 3).

For *Central*, both deer and wild pig hunt levels were associated with increased condor blood lead, but their effects differed by hunt type and time period (Fig. 3a). Specifically, we found a decline in lead levels after the 2019 ban (negative *Post2019Ban* effect) if accounting for trends in hunting, behavior, and other model covariates. Coupled with a stable positive deer hunt effect through our time series, this negative *Post2019Ban* effect is consistent with a decrease but not

cessation in lead ammunition use by deer hunters after the 2019 ban, similar to effects of ammunition regulation observed in Europe[46]. In contrast, the positive pig hunt effect steadily increased, tripling after 2008 and more than doubling again after 2019 (Supplementary Table 3a). The escalating effect of pig hunting on *Central* lead levels suggests that either the amount of lead per harvested wild pig has increased sevenfold, which is implausible, or harvest has increasingly exceeded reported tags. One potential source of pig harvest underreporting is wild pig culling, for which no robust data exist over the full time span of our analysis (Supplementary Fig. 2g, h). However, available evidence suggests wild pig populations are growing and expanding their range in California[47,48], likely leading to greater control efforts (Supplementary Fig. 4).

For *Southern*, where shifting condor behavior predicted essentially all observed increases in condor blood lead, we nonetheless detected clear and changing influences of hunting tied to the lead ammunition bans (Fig. 3b). After the 2008 ban, predicted condor blood lead decreased if accounting for trends in hunting, behavior, and other model covariates (i.e., negative *Post2008Ban* effect). This negative *Post2008Ban* effect is consistent with decreased lead-based ammunition use for hunting generally. We found an additional drop in lead exposure attributable to changes in the deer hunt after the 2019 ban (i.e., negative *DeerHunt:Post2019Ban* effect), such that higher deer hunt levels were associated with lower condor blood lead. Thus, our results indicate that after 2019, deer hunting actually benefited condors in *Southern*, suggesting that deer hunters produced enough lead-free gut piles to offset lead-contaminated sources. Conversely, wild pig hunt levels were associated with increased condor blood lead after 2019, highlighting pig hunting as an emerging source of lead for *Southern* (Supplementary Table 3b).

Non-target mortality from culling is a global concern[49–51], and the use of lead ammunition for culling wild pigs, although illegal in California, is likely no exception. We lacked data to evaluate direct links between condor lead risk and pig culling, but we hypothesize that it is a major driver of recent increases given the apparent ongoing range expansion of wild pigs in California[48] coupled with our findings of increased condor lead risk associated with pig hunting, correlations between pig hunt and pig cull levels, and increasing trends in the ratio of pig cull to pig hunt levels for *Bioyears* 2016–2022 (Supplementary Fig. 4).

Interestingly, both condor flocks displayed seasonal patterns in blood lead, with *SpringSummer* (Mar through Aug) blood lead lower for *Central* and higher for *Southern*. This seasonal variation may reflect temporal variation in pig culling and other shooting activities, such as ground squirrel control[52], for which we lack data, as well as spatio-temporal variance in condor space use (Supplementary Fig. 1i, j).

## Public outreach reduced lead exposure

Nonlead outreach activities were negatively associated with condor blood lead levels. The best outreach LMMs, which focused on lead exposure patterns after outreach programs were implemented, included the same age and behavioral variables that predicted blood lead levels for the full time series as well as two metrics of outreach—*Contacts* (number of people contacted) and *Boxes* (the number of boxes of nonlead ammunition distributed)—as well as interactions between outreach and both deer and wild pig hunt levels (Supplementary Table 5). Our models found that both *Contacts* and *Boxes* helped lower condor lead levels and were generally more effective when hunt levels were high (Supplementary Fig. 5).

Notably, our models indicate that high outreach levels can be instrumental in transitioning deer hunting to a beneficial activity for condors. Deer hunting switched from an activity that increased lead levels to one that decreased lead levels for *Central* at high levels of *Boxes* (Supplementary Fig. 5c). A similar pattern was seen for *Contacts* for *Southern*, where even moderate numbers of *Contacts* flipped deer

hunting from an activity that increased lead levels to one that decreased them (Supplementary Fig. 5e). In contrast, outreach effects on wild pig hunting were inconsistent. *Boxes* had a limited effect on lead exposure from pig hunting for *Central* (Supplementary Fig. 5d), where pig hunt levels are highest, and was most effective for *Southern* at low to moderate pig hunt levels (Supplementary Fig. 5h). *Contacts* were similarly effective for deer and pig hunting for *Central* (Supplementary Fig. 5a, b), but for *Southern*, *Contacts* only reduced blood lead at the highest pig hunt levels (Supplementary Fig. 5f).

## Only a few contaminated meals explain increased condor lead exposure

Together, our results show that multiple factors have influenced condor blood lead levels, including changes in wild behavior by condors and hunting behavior by humans. These factors could manifest in higher condor blood lead through two processes—increased frequency of encountering lead-contaminated carcasses and increased lead per feeding bout (i.e., meal). To estimate the rate at which condors encountered contaminated carcasses, we partitioned these processes using Poisson-Gamma hierarchical models (Supplementary Table 4) built with covariates from our best supported LMMs for lead exposure (Supplementary Table 3). While these models rely on multiple statistical assumptions (see "Methods": Models to predict rates of contaminated meal exposure), they provide a way to directly estimate encounter rates, a key factor that management interventions seek to influence.

For *Central*, we predict 0.59 contaminated meals made meaningful (>10 μg/dL)[5] contributions to measured blood lead levels for the average individual prior to 2008, increasing to 1.44 after 2019 (Supplementary Table 4a). For *Southern*, the average individual had 0.75 contaminated meals contributing >10 μg/dL to measured blood lead prior to 2008, increasing to 1.15 after 2019 (Supplementary Table 4b). Assuming these contamination events occurred over the previous 56 days of foraging (~4 half-lives of lead in condor blood)[53], annual consumption of contaminated meals increased from 3.8 prior to 2008 to 6.1 after 2008, and then to 9.4 after 2019 for *Central* and from 4.9 prior to 2008 to 6.0 after 2008 to 7.4 after 2019 for *Southern*. Our results should not be interpreted to imply that bans increased contaminated meals. Indeed, our Poisson-Gamma models made similar predictions to those from our lead exposure LMMs, showing bans are associated with decreases in lead contamination if accounting for changes in hunting and condor behavior.

For both flocks, the total numbers of and annual increases in contaminated meals over time were relatively small. While these results indicate that only modest reductions in contaminated carcass encounters are needed to counteract the observed rise in blood lead levels (Fig. 2a), they also underscore the challenges posed by even low levels of lead on the landscape, consistent with prior work[5]. We have previously shown that current rates of lead on the landscape are enough to transition condors in California from self-sustaining to conservation reliant[38]. Overall, our finding that this mortality level results from fewer than ten contaminated meals per year helps explain why lead poisoning of scavengers remains a global conservation problem[54–56].

## Implications for recovery

Despite lower sample sizes and a less direct connection between death and lead exposure compared to blood lead and lead exposure, our survival analysis found strikingly similar patterns to our blood lead analysis, with many predictors of blood lead also tightly linked to survival. Condors with high *Proffered* had higher $S_{Pb}$ and $S_{Overall}$, while *Presence* boosted $S_{Overall}$ for *Southern*, and *Coastal* increased $S_{Pb}$ for *Central* (Supplementary Table 6). Hunting showed similarly concordant patterns. Deer hunt levels increased *Southern* $S_{Pb}$ after the 2008 ban and increased *Southern* $S_{Overall}$ after the 2019 ban, consistent

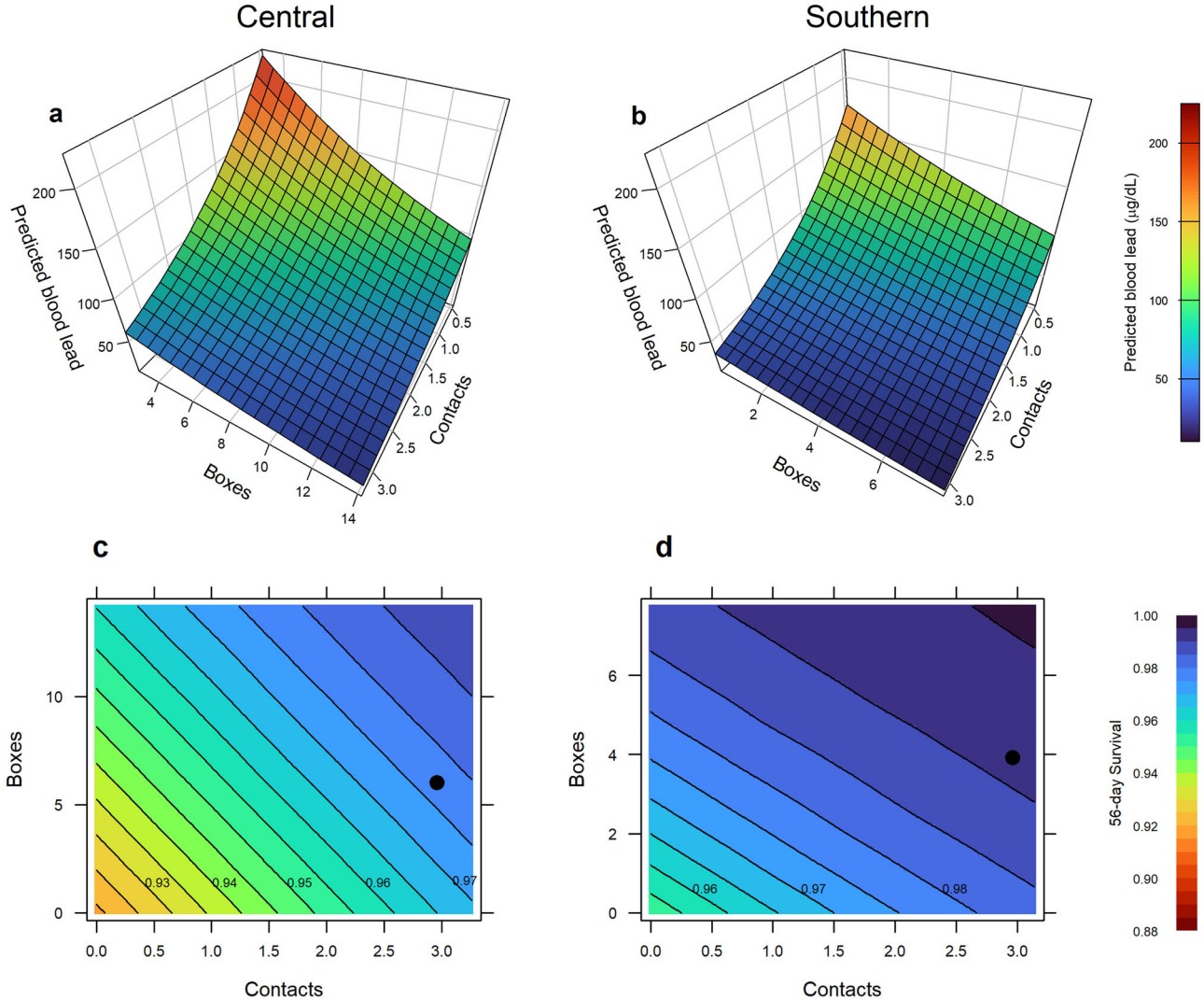

**Fig. 4 | Effects of public outreach on California condor (*Gymnogyps californianus*) lead exposure.** The number of educational contacts statewide (*Contacts*, in 1000 s, Supplementary Table 1) and boxes of nonlead ammunition distributed in counties where condors forage (*Boxes*, in 100 s, Supplementary Table 1) predicted condor blood lead levels in **a** *Central* and **b** *Southern* (see also Supplementary Fig. 5). Plotted results are for older adults (>10 years post-fledge, Supplementary Table 1) after 2019 and assume mean bimonthly peak of deer hunt, 75th percentile of pig hunt, with other model variables set at their medians. The influence of outreach on survival for **c** *Central* and **d** *Southern* was extrapolated using outreach LMMs to predict condor blood lead for a range of outreach levels at *Bioyear-* and *Season*-specific bimonthly peak deer hunt (observed seasonal means for other model variables, including pig hunt) and using Cox models to estimate 56-day survival after routine health screenings based on observed blood lead levels. Plots shown are for fall 2020 when deer and pig hunting were relatively high and include observed outreach level (black circle) (see Supplementary Figs. 6 and 7 for other years and hunt levels).

with a beneficial effect of deer hunt levels on condor blood lead levels after 2019. In contrast, wild pig hunt levels decreased both $S_{Overall}$ and $S_{Pb}$ for *Central*, with the effect strengthening after 2019, underscoring the rising lead exposure threat linked to shooting of wild pigs (Supplementary Fig. 4).

To estimate the demographic benefits of lead ammunition bans and nonlead outreach, we used our outreach models to predict the distribution of condor blood lead levels for observed hunt levels and a range of outreach levels. We also used Cox proportional hazards models to link observed blood lead levels to the probability of surviving 56 days from blood sampling. Combining these, we then estimated average 56-day survival at the peak of the deer hunt season for all outreach levels. Our models predicted that without outreach, 56-day survival during the deer hunt season would have been ~1–5% lower (Fig. 4 and Supplementary Figs. 6, 7). Given the sensitivity of population growth to survival for this extreme *K*-selected species[38], this

outreach benefit is enough to transition a declining population to a stable or increasing one.

Our analyses suggest there is potential to increase survival further with additional outreach. The beneficial effects of outreach and bans on lead exposure associated with deer hunting coupled with the rising threat associated with wild pig hunting indicate that targeting pig hunting and culling is also important for future outreach efforts.

**Broader significance**

California is currently the only U.S. state to ban lead-based ammunition for shooting wildlife, and substantial resources are expended on outreach to encourage the use of nonlead ammunition. The success of these conservation actions can serve as a bellwether for addressing lead poisoning of wildlife elsewhere, in particular for the United Kingdom[57] and European Union[58,59], where legislative restrictions on the use of lead ammunition have recently passed or are under

consideration. More generally, our findings can inform approaches to addressing the global poisoning crisis for vultures and other scavengers[35,36,60,61]. We show that despite increases in condor lead exposure and mortality, lead ammunition bans in California have been highly effective, and remain so, but countervailing forces have masked this efficacy. These forces include increased expression of wild behaviors by condors and increased shooting of wild pigs (Supplementary Figs. 2e–h and 4), both of which increase exposure of condors to lead-contaminated carcasses on the landscape. We also found that outreach activities have been highly effective in reducing lead exposure from hunting, especially for deer—the most popular group of big game species in North America and Europe. Even though many species conservation plans include public outreach components, quantifying the efficacy of outreach is typically extremely difficult and most analyses are necessarily built on implicit assumptions about these linkages[30]. We were able to leverage detailed, long-term data sets to show that for condors, a classic slow life history species, legislation and outreach played a clear, measurable role in reducing lead exposure and thus increased survival.

While many species are threatened by contaminant exposure, few are as well-studied as condors, and this wealth of data supported unique insights on recovery dynamics and the effectiveness of actions taken to address lead poisoning, an issue of global conservation concern[49,54,55,62–65]. We show that shifting behaviors and escalating threats can counteract even highly effective actions, leading to a situation, in which "…it takes all the running you can do, to keep in the same place"[66]. Organizational theory widely recognizes this as a Red Queen dynamic, which occurs when system changes require increasing effort and innovation to sustain the same level of success[67–69], the exact scenario our results have uncovered for condor recovery actions implemented amidst changing ecological conditions. Our findings underscore the importance of being alert for Red Queen-like dynamics when evaluating conservation measures in complex settings where changing behaviors and threats can mask management effectiveness, with harmful outcomes if successful policies and actions were to be abandoned.

## Methods

California condor (*Gymnogyps californianus*, condor) behavioral data and biological samples (blood, feathers) were collected and analyzed according to all relevant ethical regulations and permit authorizations. Specifically, personnel affiliated with the California Condor Recovery Program collected behavioral data and biological samples as part of routine monitoring of condors in California, USA and Baja California, Mexico under appropriate state and federal permits as follows: Hopper Mountain National Wildlife Refuge Complex Federal permit: ES108507 (previously 02360A HMNWRC), California state permit: 2081a-2014-047-00; Ventana Wildlife Society Federal permit: TE 026659, California state permit: 2081a-2014-054-00; Pinnacles National Park Federal permit: ES157291-3, California state permit: 2081a-2014-048-00, Institutional Animal Care and Use Committee (IACUC) through the National Park Service Biological Resources Division: CA_PINN_Welch_Condor_2021.A3; Parque Nacional Sierra de San Pedro Mártir: Federal authorization from the Secretaría de Medio Ambiente y Recursos Naturales (SEMARNAT) of Mexico. California condor biological samples were received and analyzed with the University of California, Santa Cruz IACUC approval under the following protocols: Smitd0712, Finkm1102, FINKM1307, FINKM1607, Finkm1907, Finkm2207dn.

### Natural history and management relevant to data
California condors are critically endangered. Their life history strategy is characterized by a long lifespan and very low reproductive rates[70]. Following complete extirpation in the wild in 1987, captive releases were initiated in 1992 to re-establish wild flocks. During the period of our analyses, Sep 1996–Jun 2023, condors occurred in two flocks in

California (the Central flock, hereafter *Central* and the Southern flock, hereafter *Southern)* and in one flock in Baja California, Mexico (hereafter *Baja*) (Fig. 1 and Supplementary Fig. 1). The primary source of mortality for free-flying condors in *Central* and *Southern* flocks, Sep 1996–Jun 2023 was lead toxicosis (see also refs. 38,40).

Flock managers provisioned condors with proffered carcasses to facilitate capture and monitoring. Flock managers conducted annual or twice-annual trap-ups, attempting to capture all free-flying birds to affix transmitters (see "Methods": Condor space use), vaccinate, and perform health checks, including drawing blood for lead measurements. Condors with elevated blood lead, exhibiting clinical symptoms of lead toxicosis, or other medical need, may have received treatment in zoo facilities, and condors may have been held captive in field pens for extended time periods for a variety of medical or management needs.

For all analyses, we used an annual increment starting at approximate fledging time (i.e., *Bioyear*, Sep through Aug). We omitted data from *Bioyear* 1992–1995 when few condors were present in the wild and limited blood lead measurements (<10) were taken. We also omitted Jul–Aug 2023 because no hunt data were available for these months.

### Condor daily data
All free-flying condors and offspring that survived to fledging were known, representing a complete population census. Condors wore uniquely identifiable wing tags and a majority wore functional telemetry tags (>81% *Central*, >69% *Southern*). On a near-daily basis ($\bar{x} = 346 \pm 5.6$ SE days year$^{-1}$), biologists attempted to locate each condor's telemetry signal or visually sight each condor from tracking locations or on remotely triggered cameras, including at proffered feeding stations. We used three indices of foraging and ranging to summarize behavior: the proportion of free-flying days in the previous 180 days that an individual was (1) observed feeding at proffered carcasses (*Proffered*), (2) detected via telemetry or visual observation, excluding proffered feeding observations (*Presence*), and (3) observed visually in the coastal region (*Coastal*, for *Central* birds only) (Supplementary Table 1).

### Condor blood lead levels
Condor biologists collected whole blood samples from the metatarsal vein for routine health monitoring as well as suspected lead poisoning events. Blood samples were sent to a certified commercial laboratory (ANTECH Diagnostics, Louisiana Animal Disease Diagnostic Laboratory, and/or California Animal Health and Food Safety Laboratory, University of California, Davis) for determination of lead concentrations (*BloodLead*, Supplementary Table 1). If no lab value was available, we used lead values from LeadCare or Lead Care II field testers (<15% of samples).

To ensure independence, we restricted blood samples to those >56 days from a previous sample, with one exception: if a sample was taken 14–56 days from a previous sample but was >10 μg/dL higher, this was considered an independent exposure event and included. If a sample was taken <14 days from a previous sample and was higher, we used the second sample, assuming it captured rising lead levels from the same exposure event. We excluded samples if an individual was in veterinary care >3 days or field pens >21 days. Applying these rules yielded 1421 samples for 188 condors over 26 years for *Central*, 1758 blood lead samples for 179 condors over 27 years for *Southern*, and 446 samples for 66 condors over 20 years for *Baja*. Laboratory detection limits were reported up to 5 μg/dL; thus, we set all blood lead values ≤5 μg/dL to 2.5 μg/dL.

### Condor space use
Managers fit condors with GPS patagial telemetry devices (Microwave Telemetry, Inc., Columbia, Maryland, USA or Cellular Tracking

Technologies, Rio Grande, New Jersey, USA) that transmit data via the Global System for Mobile Communications. Telemetry units collected GPS data at intervals from 0.5 to 60 min, including location, date, time, altitude above an ellipsoid or above mean sea level, fix quality (2D or 3D), and horizontal dilution of precision (HDOP), and for some units, vertical dilution of precision (VDOP). We subsampled high-frequency locations to 15-min intervals. To limit uncertainty and measurement error, we removed locations with 2D fix quality, HDOP or VDOP values >10, reported altitudes >4000 m, and altitudes above ground level ≤−50 m (defined relative to a 30-m resolution digital elevation model)[71,72].

We defined likely foraging locations as dawn to dusk ground locations for free-flying condors with speed <5 kph and altitude above ground level <50 m. We collected foraging locations from 192 condors (52 *Central*, 140 *Southern*) between July 2013 and January 2024 ($\bar{x} \pm$ SD: 754 ± 559 days bird$^{-1}$ for *Central*, 1301 ± 896 days bird$^{-1}$ for *Southern*), or 4,794,548 foraging locations total (725,987 and 13,961 ± 11,916 locations bird$^{-1}$ for *Central*, 4,068,561 and 29,061 ± 19,238 locations bird$^{-1}$ for *Southern*). To identify high-use foraging counties, we summarized the proportional foraging use of counties from all GPS tagged condors by *Bioyear* and season with ArcGIS Pro 3.3.1.

## Hunting and culling levels
We indexed deer and wild pig hunt levels based on monthly hunter tag reports (excluding archery) for high-use foraging counties (see "Methods": Condor space use) (Supplementary Fig. 2). A change in regulations resulted in an increase in reporting for deer tags after 2014. To account for this change, we assumed that the overall mean hunt level was constant from 2007 onward based on the numbers of deer tags sold statewide (Supplementary Fig. 2a, b) and used the ratio of statewide tag reports for two 7-year periods, 2007–2014 vs. 2015–2022, as a correction factor. Specifically, we increased monthly tag report counts by 1.53 for all years before 2015 to account for underreporting relative to 2015–2022 (See Supplementary Fig. 2a). We summarized data as 2-month running averages (Supplementary Fig. 2c, d), using hunting in the current and previous month to predict condor blood lead in the current month.

Culling of wild pigs occurs mainly on private land, where pigs can cause widespread property damage. Permitting and reporting regulations have changed over the past 20 years, such that reliable records of cull levels are unavailable before 2017 (Supplementary Fig. 2g, h) preventing analysis of pig culling in condor lead risk models. However, we calculated correlations between pig hunting and culling levels (corr function in R)[73] and performed simple linear regressions to assess trends in the relationship between PigCull/PigHunt (lm function in R) for *Bioyears* 2016–2022 (Supplementary Fig. 4). *Bioyear* 2016 only included data from Jan 2017 to Aug 2017 (vs Sep 2016 to Aug 2017) for these analyses due to lack of robust culling data for 2016 and earlier.

## Public outreach levels
We used data from public outreach initiated in 2008 and conducted by staff affiliated with Pinnacles National Park, Ventana Wildlife Society, and Institute for Wildlife Studies. Outreach personnel provided information on lead-based ammunition bans, environmental and health effects of lead-based ammunition, and effective alternative ammunition (e.g., nonlead, also referred to as lead-free); outreach also included distribution of nonlead ammunition starting in 2012 for *Central* and in 2014 for *Southern*. Outreach occurred at community events, such as shooting demonstrations and meetings of hunting organizations, as well as direct outreach to landowners. We summarized outreach levels as the number of people contacted (*Contacts*) and the number of boxes of ammunition distributed (*Boxes*). To maximize differentiation between *Contacts* and *Boxes*, we excluded from *Contacts* those that included the receipt of ammunition. We summarized flock-specific outreach occurring in high-use foraging counties as well

as statewide (Supplementary Table 1 and Supplementary Figs. 1, 2). Outreach metrics were compiled for calendar year and used to predict the *Bioyear* starting in Sep (e.g., outreach Jan–Dec 2020 was used to predict *BloodLead* Sep 2020–Aug 2021). Available outreach data ended after *Bioyear* 2021. Although we do not have data that directly measures the amount of nonlead ammunition used per contact or box of ammunition distributed, we assume that nonlead ammunition use is positively related to numbers of contacts and numbers of boxes of ammunition distributed.

## Condor feather lead levels
Feathers were randomly collected (i.e., not associated with known lead exposure events) between 1997 and 2022 and processed and analyzed using established methods and trace metal clean techniques under HEPA-filtered air laboratory conditions[39,41]. Lead concentrations were determined by inductively coupled plasma mass spectrometry (ICP-MS, Thermo Element XR high-resolution or X-Series II quadrupole), measuring masses of 206Pb, 207Pb, and 208Pb with 205Tl added as an internal standard at the University of California Santa Cruz Plasma Analytical Laboratory (RRID:SCR_021925). Each individual section of feather analyzed equaled ~4–5 days of growth[41], and at least 3 sections were analyzed per feather. The section with the highest lead concentration (peak feather lead) was used to represent the maximum lead concentration measured per feather[63]. To compare with blood lead data, we estimated blood lead (μg/dL) from feather lead (μg/g) using a ratio of 20:1[39,41].

## Models to predict lead exposure
To identify predictors of condor blood lead levels, we built linear mixed-effects models (LMMs) for *Central* and *Southern* separately using the lmer function in R[73] package lme4[74]. We used natural-log-transformed *BloodLead (lnPb)* as a response variable and considered factors that may influence individual likelihood of encountering lead, including whether *BloodLead* was measured due to a suspected lead exposure event (*Targeted*), condor age and release time, condor behaviors (i.e., *Proffered, Presence, Coastal*), and the proportion of the previous 6 months that a condor was free-flying (*FreeFly*), as well as factors that may influence the amount of lead in the environment (e.g., season, deer and wild pig hunt tags returned or *DeerHunt* and *PigHunt*, and binary variables for the time after the 2008 and 2019 lead bans, i.e., *Post2008Ban, Post2019Ban*) (Supplementary Table 1). Note that the *Post2008Ban* effect is cumulative with the *Post2019Ban* effect in the time period from 2019 onward. In addition to overall deer and pig hunting variables, we also considered time-period specific hunt effects (e.g., *PigHunt:Post2008Ban*, Supplementary Table 1). To test for effects of outreach activities on condor blood lead exposure, we ran separate analyses using only the years for which we had data for both *Contacts* and *Boxes* (see "Methods": Public outreach levels, Supplementary Table 1, Supplementary Fig. 2 and Supplementary Note 1).

We used model AICc to guide model selection, comparing results for all possible combinations of variables and interactions selected a priori via the dredge function in the R package MuMIn[75]. We also calculated and plotted (Supplementary Fig. 3) several measures of variable importance to guide our interpretation of the drivers of lead exposure. Specifically, we calculated the sum of AICc weights for all models that contained each predictor variable (MuMIn, sw function), a widely used method, but one that has recently been both criticized as imprecise and also defended[76–78]. We also ran models using standardized coefficients (centered and multiplied by the partial standard deviation, which adjusts for multicollinearity among variables, using the beta = "partial.sd" option in the dredge function) and calculated model-averaged coefficients for these standardized variables (MuMIn, model.avg function). We used these coefficients to calculate variable importance as the ratio of the absolute value of the model-averaged coefficient to the maximum model-averaged coefficient[76,77], using AICc

weights for all models ("full") and only the subset of models containing the variable ("subset"). Finally, in our model selection tables, we indicate models that have higher ranked models nested within them and thus contain parameters that are likely to be uninformative[79,80] (MuMIn, nested function). Finally, we visually depicted the relative importance of different classes of variables by plotting annual variance as categorical year effects for successively more complex models nested within the best-supported model (Fig. 3). We included condor *ID*, a repeated measure, and *Bioyear* as crossed random effects in all models, except when depicting annual changes (i.e., Fig. 3 omits *Bioyear* as a random effect). We made visual and statistical checks of model assumptions (normality of residuals and random effects, linearity, homogeneity of variance, multicollinearity) using the R package performance[81] as well as partial residuals plots using the R package ggeffects[82]. We report *p*-values from the R package lmerTest[83] and marginal and conditional $R^2$ values[42] from the MuMIn r.squaredGLMM function.

## Models to predict survival

We built known fate survival models for all post-fledge condors using the R[73] package rMark[84] interface for program MARK[85], analyzing data for both flocks together. We used these models to identify factors that predicted survival, considering the same classes of predictor variables that were used in lead exposure analyses, including condor behaviors, years bans were in effect, and hunt levels. We separately analyzed overall survival ($S_{Overall}$) and survival of lead hazards ($S_{Pb}$), for which we censored individuals that died of causes other than lead from the analysis at the time of death (i.e., removed alive at time of death)[38]. Annual records for individuals were censored if they spent more than 365 consecutive days in captivity. We also censored deaths due to a catastrophic wildfire that occurred in Aug 2020 in *Central*.

Our model-building strategy consisted of first identifying the best functional forms for changes in survival with age and time since release. Previous analyses documented lower survival for newly fledged and newly released birds[38]. We also tested for differences by other age groups and by time (i.e., early time periods when flocks were small and managers were refining protocols). Finally, we used the best model for variation by time, age, and release time and compared support for models with behavioral variables, lead bans, and hunt levels, with and without interactions by flock. We used AICc to guide model selection. As with LMMs of *BloodLead*, we considered time-period-specific hunt effects and flock-specific hunt and behavioral effects in addition to whole time period and whole population effects (e.g., *PigHunt:Post2008Ban*, *CECA:Proffered*, see Supplementary Table 1 and Supplementary Note 2).

We also built a simple known fate survival model for all post-fledge condors in *Central*, *Southern*, and *Baja* using rMark[84] to estimate overall survival ($S_{Overall}$) by flock, age, and sex to parameterize a female-only matrix model (Supplementary Table 2 and Supplementary Note 3).

## Models to predict rates of contaminated meal exposure

To estimate the number of lead-contaminated meals contributing to the lead present in a blood sample, we fit Poisson-Gamma regression models[86,87] to the same data used in LMM analyses of factors influencing blood lead levels (see "Methods": Models to predict lead exposure). Poisson-Gamma models estimate the number of encounters contributing to a summed outcome (Poisson portion of model) and the distribution of contributions from each encounter (Gamma portion). These processes are often fit using Tweedie models[88,89], but this condensed form does not allow the flexibility to model different effects in the Poisson and Gamma processes, so we fit models directly using MCMC methods via the R[73] package JAGS[90].

For each flock, we fit a Poisson-Gamma model using the covariates in the best-supported LMM (Supplementary Table 3). We expected

that most covariates would influence encounter rates, rather than mean lead per meal, and thus we included all covariates in the Poisson portion but included *Targeted* in both Poisson and Gamma portions (Supplementary Table 4).

Poisson-Gamma models are predicated on several statistical assumptions that are difficult to directly test with our data. Most notably among these is that encounters are independent and well-described by the Poisson distribution, and that individual encounters contribute to the summed observed blood lead levels seen in a sample with Gamma distributed values. Both these assumptions are reasonable, but without exceptionally detailed data on individual contaminated meals, they are impossible to test. One process we could not model separately and that must be acknowledged is the attenuation of lead present in the blood following feeding on a contaminated carcass. The total blood lead observed at sampling is a combination of lead from different contaminated meals, and the amount of these contributions is determined both by the quantity of lead ingested from a meal and how long ago the meal occurred as well as the "body burden" of lead in organs[91]. The mean lead per meal estimated through the Gamma portion of the model necessarily combines these processes into the mean and variance in the contributions per meal observed at sampling.

We assumed that predictors influenced the mean of the Poisson and Gamma distributions, but that the shape parameter of the Gamma distribution is constant. To achieve convergence, we standardized hunt variables by dividing by their maximum values. Finally, blood lead levels <5 μg/dL were fit using the cumulative distribution function in our JAGS code.

We modeled the mean encounter rate and mean lead per meal using log link functions and estimated parameters using MCMC fitting in JAGS, implemented through the run.jags function in the R package runjags[92]. All coefficient parameters for both the Poisson and Gamma were given uninformative priors, sampling from uniform distributions with bounds {−100, 100} while zeta, the transformed shape parameter for the Gamma, (shape = (2 − zeta)/(zeta − 1)) was given a uniform prior of {1, 2}. We achieved good diagnostics for these model fits using 3 chains, thinning = 3, and 40,000 samples.

Following model fitting, we used the results to predict how the encounter rate of lead-contaminated carcasses changed over time (Supplementary Table 4). Specifically, we estimated the carcass contamination rate for adult condors for the time periods before the 2008 ban, after the 2008 ban, and after the 2019 ban, setting time-varying covariates to their means for each time period and *Targeted* to 0. Because our models predict that most encounters contribute extremely small amounts of lead to blood samples, we estimated the number of meals contributing ≥10 μg/dL (a recognized blood lead exposure threshold for condors)[39] to sampled blood lead levels, or essentially the number of meals making meaningful contributions to observed blood lead. We assumed that contamination events contributing meaningfully to measured blood lead occurred over the previous 56 days of foraging (~4 half-lives of lead in condor blood)[53] and used this value to extrapolate to annual contaminated meals by multiplying by 6.5 (56-day periods in a year).

## Models to estimate population growth rate

We built a female-only age- and stage-structured matrix model for California condors, with a fledging time census. To parameterize this matrix and estimate population growth rates, we used our survival models to estimate overall survival for different classes of post-fledge females. Based on previous work[93], we tested for differences by flock, age (*FirstYear*: fledge to 1 year post-fledge, *Juvenile*: 1–6 years post-fledge, *YoungAdult*: 6–10 years post-fledge, and *OlderAdult*: ≥10 years post-fledge), and release time (e.g., released <1 year) (see "Methods": Models to predict survival). We estimated fertility, defining breeding success as fledging a chick. The first successful fledging event

transitioned an individual to recruited breeder status. Breeding success differed by individual age, breeding history in the previous year, free-flying days in the previous 180, and whether the event occurred in the early breeding years[94] (Supplementary Tables 1 and 15). Breeders that successfully fledged a chick in the previous year (*SuccessfulBreeder, SB*) were least likely to fledge a chick, while breeders that failed to fledge a chick in the previous year (*FailedBreeder, FB*) were most likely to fledge a chick. Unrecruited breeders (i.e., *NonBreeder, NB*) had intermediate breeding success. We used data for *Central, Southern,* and *Baja* to estimate age($i$)-specific probabilities of fledging a chick (*Succ*) for *NB, SB,* and *FB*, based on supported age differences. *PreBreeders (PB)* were females <age 5, the first age at which breeding is ever observed. We assumed a sex ratio (*female*) of 0.5 for fledged chicks.

Based on these analyses, we built a 16-stage population matrix model (Supplementary Tables 2, 7, 8 and Supplementary Note 3), with *Succ* based on previous year's breeding history incorporated into the breeder transitions, and the reproduction term on the first row of the matrix consisting of the probability of a fledged chick being female and surviving a full year from fledging (*female* × $S_{FirstYear}$)[2,38]. To identify predictors of *Succ*, we built generalized LMMs using the glmer function in the R[73] package lme4[74] with a binomial response variable and condor *ID* as a random effect and obtained *p*-values using the R package lmerTest[83].

### Effects of outreach on survival
We used a multistep process to predict the effects of outreach on survival. (1) We established a relationship between *lnPb* and survival, using Cox proportional hazards models (coxph function in the R[73] package survival[95]) to estimate the probability of surviving 56 days ($S_{56}$) after a measured blood lead value, with *lnPb* as a predictor variable. We censored *Targeted* samples from these analyses as they are often from individuals that exhibit clinical symptoms indicating health issues, including those that may be unrelated to lead. We also censored *Central* individuals killed by an Aug 2020 catastrophic wildfire. (2) We predicted flock-wide mean *lnPb* based on our outreach LMMs for a range of outreach levels (*Boxes* and *Contacts*) with *Targeted* set to 0, *PigHunt* set to seasonal means, *DeerHunt* set to peak bimonthly levels, and other values set to their year-specific means. (3) We used our Cox models (from step 1) to estimate 56-day survival for the predicted mean *lnPb* values associated with differing outreach levels (from step 2). Given the nonlinear relationship between *lnPb* and survival, we computed a probability density function for *lnPb* values based on the residual variance from outreach LMMs. We then used normalized probabilities for each *lnPb* level to calculate the weighted-average $S_{56}$ for Cox-model-estimated survival rates.

Finally, we established relationships between survival and outreach. We used simple linear regression models to predict $S_{56}$ from step 3 based on the *Boxes* and *Contacts* ($S_{56}$ = *Contacts* x *Boxes*) used to estimate *lnPb* from step 2 and plotted 3D relationships using the levelplot function in the R[73] package lattice[96]. Because the relationship between $S_{56}$ and *Boxes* and *Contacts* changes depending on the levels of other predictors in our outreach LMMs, especially hunt levels, which interact with outreach, we evaluated these relationships under a range of observed hunt levels for many different years and set the other model variables to their year-specific values.

### Reporting summary
Further information on research design is available in the Nature Portfolio Reporting Summary linked to this article.

## Data availability
All data required to replicate analyses and produce figures presented in this study are provided in Supplementary Information as Supplementary Data 1. These include data on condor behavior and blood lead levels as well as data on public outreach effort and deer and pig tag returns. The spatial data used to identify high-use foraging counties for the Southern California flock were collected by the U.S. Fish and Wildlife Service and are publicly available on ScienceBase https://www.sciencebase.gov/catalog/item/546f5ec5e4b0b935bc7586e0 while those for the Central California flock were collected by Ventana Wildlife Society and Pinnacles National Park and some are under restricted access due to endangered species management policy. Qualified researchers may obtain these data by contacting the corresponding author and agreeing to data sharing restrictions. Data from the California Protected Areas Database are available at https://data.cnra.ca.gov/dataset/california-protected-areas-database. Source data are provided with this paper.

## Code availability
The code needed to reproduce the analyses and generate the figures in this study are included in Supplementary Data 1.

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

## Acknowledgements

We thank Don Smith, Juan J. Vargas Velazco, J. Hiram Licona Hdz., Alejandra Arguelles C. and LightHawk for intellectual contributions and data collection efforts. We also thank the Institute for Wildlife Studies for outreach data and Dan Skalos and California Department of Fish and Wildlife for hunting and culling data. The findings and conclusions in this article are those of the authors and do not necessarily represent the views of the U.S. Fish and Wildlife Service. Any use of trade, firm, or product names is for descriptive purposes only and does not imply endorsement by the U.S. Government. Funding: U.S. Department of the Interior awards F21AP01583, F15AC01140, F11AC00781 (MEF) and National Park Service awards P22AC02283, P19AC01001, P15AC00724 (MEF).

## Author contributions

V.J.B., M.E.F., and D.F.D. conceived of the project, developed the methodology, and wrote the original draft. V.J.B., D.F.D., M.E.F., S.A.P., and M.A.B. curated and analyzed the data. M.E.F., S.K., A.W., and R.W. acquired funding. M.E.F., A.W., R.W., D.R., L.J.B., M.S., A.P., J.B., S.K., M.C.P.P., and N.V. administered data collection and provided resources. All authors reviewed and edited the final manuscript.

## Competing interests

The authors declare no competing interests.
