## [Transparent Peer Review file · Nature Communications]

Behavioral shifts mask the success of legislation and outreach for endangered species recovery

Corresponding Author: Dr Myra Finkelstein

Version 0:

Reviewer comments:

Reviewer #1

(Remarks to the Author)

The central thesis of this paper is that changing conditions can complicate evaluating the impact of management actions taken to recover endangered species. This thesis is of more general significance than it might at first appear as it applies more broadly to species that are declining but not yet endangered such as the tipping point species that are the focus of the Road to Recovery avian conservation program. An excellent introduction provides a well-developed conceptual framework for exploring this topic. Key points include the particular difficulties, given this reality, of evaluating the impact of management actions that work on long time scales, especially for species with slow life histories, and justifying costly, controversial management actions. Changes in the behavior of target species and humans and increasing reliance on human-modified landscapes are highlighted as key components of changing conditions. A secondary theme is the role of outreach in achieving conservation success.

The study species, the California Condor, is ideal for this study. First, there is overwhelming evidence that the cause of the decline of this species is reduced survival due lead poisoning resulting from ingestion of lead ammunition while feeding on carcasses and that the primary objective of conservation actions must be to reduce exposure to this mortality factor. Second, condors have a slow life history that severely limits the pace of their recovery and the key management action taken to promote recovery, banning lead ammunition, is highly controversial. Third, the authors have sufficient relevant data to tackle the task they have set for themselves. The extent of knowledge of every individual in the population is unique. What every condor was doing and where it was doing it is known for virtually every day for every condor since the beginning of their reintroduction to the wild. Specifically, the authors have extensive data on condor foraging behavior, survival rates, causes of mortality based on necropsies, and blood lead levels. They also have extensive data on factors likely to affect exposure to lead ammunition such as deer and pig hunting and outreach activities.

The results are compelling. The authors provide strong evidence that blood lead levels have increased and in parallel survival has decreased over time in two declining California populations. Interestingly, they also document that exposure to lead and mortality have decreased over time in an increasing population in Mexico. This finding provides further evidence that exposure to lead ammunition limits condor recovery, and identifies the Mexican population as a potential reference or control site. Modeling indicates that birds that feed more on proffered food and stay more in the bait area have lower blood levels than those that range more widely and thus feed more on carcasses they discover. (Feeding on the coast on marine mammal carcasses rather than inland on carcasses of other animals has the same effect in one population.) Model results also suggest that after lead was banned deer hunting became associated with reduced rather than higher blood lead levels, suggesting that fewer hunters are using lead ammunition, whereas increases in pig hunting over time are associated with increases in blood lead levels. Finally, the authors provide modeling evidence of a positive relationship between outreach activities and reductions in blood lead levels that appears to be due to its impact on lead exposure from deer hunting. For the most part, results for survival mirror those for blood lead levels, that is, factors that reduce blood lead levels are also associated with increased survival.

The authors conclude from their results that increases in "wild" foraging (as opposed to reliance on proffered carcasses) and in pig hunting have driven the observed increases in blood lead levels and lead-related mortalities, and that these changes mask the positive effects of lead bans and outreach activities on exposure to lead ammunition, as when the effects of these negative factors are accounted for blood lead levels are lower than previously. I think this is a fair and interesting interpretation of the body of evidence presented.

The demonstration of a positive effect of outreach also is an important result. Previous work has shown that outreach increases awareness of conservation issues and needed management actions, but this increased awareness generally cannot be linked to conservation success or changes in human behavior. The results of this study make both of these links. Nevertheless, I think the authors sometimes highlight this conclusion at the expense of their conclusion about the effect of the bans on lead ammunition. The latter action is the more controversial one, hence evidence of its effectiveness is critical and should be highlighted more in my opinion.

Clarity of presentation is generally good, but I found the information provided about pig hunting sufficiently lacking to seriously compromise my ability to interpret results. What is lacking is information about use of lead ammunition in pig hunting. Is the use of lead ammunition legal for pig hunting? If not, what can account for pig hunting leading to more exposure to lead than deer hunting? What is the difference between pig hunting and feral pig depredation? If depredation refers to killing of pigs by non-human predators, why would that result in exposure to lead? My other criticism of the presentation is that many of the figures are very complex, requiring the reader (at least this reader) to study them for a while in order to understand them. Figure 3 is particularly difficult to understand, and the description of the relevant modeling in the methods is not detailed enough to be helpful in understanding this figure. What is a "residual trend"? (I also think there is a mistake in the legend, a (c) that should be and (a).

My other criticisms of the paper concern the modeling analyses. In the modeling of blood lead levels (and survival to a lesser extent), a huge number of models have very similar AIC values. The authors need to provide more details in the methods about how they handled this in the context of uninformative variables, model averaging and coefficient estimates. The modeling generally is firmly grounded in data and evidence. The one exception is the modeling of contaminated meal exposure. There are a number of unsupported assumptions in this analysis such that I found the validity of the results questionable. This analysis seems to me somewhat of a tangent in that it attempts to provide a level of detail about lead exposure that is not necessary to the major conclusions of the study. Because the results essentially duplicate the blood lead level results, I found them uninformative. I suggest deleting this analysis.

(Remarks on code availability)

Reviewer #2

(Remarks to the Author)

This manuscript makes use of extensive datasets to explore why recovery efforts for the California Condor in the U.S. appear to be stalled due to increases in lead poisoning, despite a ban on lead ammunition in California and extensive outreach/education efforts about the use of non-lead ammunition. The authors show that an increase in more natural behaviors by condors (i.e., feeding from non-subsidized food, and moving over large distances) and an increase in hunting pressure together mask the success of lead ammunition bans and outreach campaigns.

The authors brought a myriad of data to address their study objectives, and I appreciated the depth they went to look at this issue. I think the authors could strengthen the MS by considering changes to the framing of the study and broadening the implications of their study beyond the California Condor. I expand on these and other comments below.

1. The idea of describing the study as one that is illustrative of Red Queen dynamics is interesting, but given *Nature Communications* is a multi-disciplinary journal I suspect it will not resonate with a broader audience. What's more, the authors do a poor job of explaining what is meant by Red Queen dynamics by largely focusing on a quote from Alice in Wonderland, which is bound to be confusing to some. Therefore, I suggest dropping the Red Queen piece and instead focus on the idea that species recovery can be masked by temporal changes in behavior, which I expect to resonate with readers much more than Red Queen dynamics.

2. The MS has an extensive analysis of condor data, but in places it is rather limited beyond this species. Although the Introduction provides information that goes beyond condors, the Discussion does not, and instead is almost exclusively focused on condors. Towards the end of the MS the authors try to argue that their findings for condors likely apply to other systems, but they also note that the data available for this species is exceptional and that few other species have such a wealth of data available. That makes me question whether the impact of this study is restricted mostly to condors, or if it goes further and can inform us about other systems. For broad readership in a multidisciplinary journal it has to be the latter, and therefore I suggest the authors should make the Discussion more robust in this regard.

3. The authors state that one of the key findings from the study is that outreach activities have been highly effective at reducing lead exposure from hunting, but the data that are used are observational (i.e., the number of people contacted at outreach events, and the number of non-lead ammunition distributed) and require untestable assumptions. For example, increases in the distribution of non-lead ammunition does not necessarily mean that such ammunition was used for hunting the focal species (deer or pigs) that often contaminate condors with lead. Therefore, my view is that more caution is needed when drawing conclusions about the effect of outreach activities. Additionally, because there are ostensibly no detailed data on human behaviors related to these outreach efforts (e.g., how often distributed ammo was used for hunting pigs/deer by those who received it), the authors would do well to highlight the need for new studies focused on this topic to bring a more comprehensive understanding of how outreach activities impact condor recovery.

4. The authors used the Baja population as a control for the U.S. populations, but it was noteworthy that this population has $\lambda < 1$ in Fig. 2b despite lead ammunition not being a concern. It might seem a bit beyond the scope of this MS, but I think readers will be curious about why that's the case, and whether the factor(s) in Baja that prevent recovery may also be present in the U.S. population and have the potential to interact with lead poisoning. This is important because even if lead poisoning is reduced through recovery efforts in the U.S., addressing the other factor(s) are likely to be as important – or more important – for effective long-term recovery of the species.

(Remarks on code availability)

Reviewer #3

(Remarks to the Author)

Thanks for the opportunity to review this manuscript. Overall, I think the RQs are well articulated, the methods are sound, and the findings are interesting and (mostly) well contextualized in the literature. I have only two substantive concerns:

1) The increasing "wilding" behavior of breeding program condors has been studied in other contexts and you might do well to engage with new literature on the species, even look to the social science and humanities literature on condors. I might suggest Farnsworth, J. S. 2015. The Condor Question Revisited as one example. I think you could contextualize and situate your findings a bit more in the rich world of condor research.

2) I suggest you craft a paragraph addressing all threats to your internal validity (of your data metrics and operational definition). You could say a bit more, too, regarding generalizability but the first part is more important.

(Remarks on code availability)

Version 1:

Reviewer comments:

Reviewer #1

(Remarks to the Author)

I appreciate the serious consideration the authors gave my previous comments. The revisions the authors made satisfied my previous concerns. In the one case where the authors disagreed with a change I suggested (deleting the model analysis of contaminated meat exposure) the authors make a convincing case for retaining this analysis and clarify the strengths and weaknesses of this analysis in their revision. I have no criticisms of the revised version of this paper.

(Remarks on code availability)

Reviewer #3

(Remarks to the Author)

Overall, this is well structured. I believe you need to improve the way you discuss threats to internal validity and threats to external validity in your discussion/conclusion.

(Remarks on code availability)

We would like to extend our sincere gratitude to the three anonymous reviewers for their thoughtful and constructive comments and suggestions. We strongly believe that our manuscript is much improved based on our revisions in response to these reviews.

We provide a detailed numbered response to each comment below and submitted a version of our manuscript with our revisions marked using the track changes feature in Word. Provided line numbers correspond to our submitted manuscript with changes tracked.

DETAILED RESPONSE TO REVIEWER COMMENTS

REVIEWER #1 (REMARKS TO THE AUTHOR):

1.1 Comment: The central thesis of this paper is that changing conditions can complicate evaluating the impact of management actions taken to recover endangered species. This thesis is of more general significance than it might at first appear as it applies more broadly to species that are declining but not yet endangered such as the tipping point species that are the focus of the Road to Recovery avian conservation program. An excellent introduction provides a well-developed conceptual framework for exploring this topic. Key points include the particular difficulties, given this reality, of evaluating the impact of management actions that work on long time scales, especially for species with slow life histories, and justifying costly, controversial management actions. Changes in the behavior of target species and humans and increasing reliance on human-modified landscapes are highlighted as key components of changing conditions. A secondary theme is the role of outreach in achieving conservation success.

The study species, the California Condor, is ideal for this study. First, there is overwhelming evidence that the cause of the decline of this species is reduced survival due lead poisoning resulting from ingestion of lead ammunition while feeding on carcasses and that the primary objective of conservation actions must be to reduce exposure to this mortality factor. Second, condors have a slow life history that severely limits the pace of their recovery and the key management action taken to promote recovery, banning lead ammunition, is highly controversial. Third, the authors have sufficient relevant data to tackle the task they have set for themselves. The extent of knowledge of every individual in the population is unique. What every condor was doing and where it was doing it is known for virtually every day for every condor since the beginning of their reintroduction to the wild. Specifically, the authors have extensive data on condor foraging behavior, survival rates, causes of mortality based on necropsies, and blood lead levels. They also have extensive data on factors likely to affect exposure to lead ammunition such as deer and pig hunting and outreach activities.

The results are compelling. The authors provide strong evidence that blood lead levels have increased and in parallel survival has decreased over time in two declining California populations. Interestingly, they also document that exposure to lead and mortality have decreased over time in an increasing population in Mexico. This finding provides further evidence that exposure to lead ammunition limits condor recovery, and identifies the Mexican population as a potential reference or control site. Modeling indicates that birds that feed more on proffered food and stay more in the bait area have lower blood levels than those that range more widely and thus feed more on carcasses they discover. (Feeding on the coast on marine mammal carcasses rather

than inland on carcasses of other animals has the same effect in one population.) Model results also suggest that after lead was banned deer hunting became associated with reduced rather than higher blood lead levels, suggesting that fewer hunters are using lead ammunition, whereas increases in pig hunting over time are associated with increases in blood lead levels. Finally, the authors provide modeling evidence of a positive relationship between outreach activities and reductions in blood lead levels that appears to be due to its impact on lead exposure from deer hunting. For the most part, results for survival mirror those for blood lead levels, that is, factors that reduce blood lead levels are also associated with increased survival.

The authors conclude from their results that increases in “wild” foraging (as opposed to reliance on proffered carcasses) and in pig hunting have driven the observed increases in blood lead levels and lead-related mortalities, and that these changes mask the positive effects of lead bans and outreach activities on exposure to lead ammunition, as when the effects of these negative factors are accounted for blood lead levels are lower than previously. I think this is a fair and interesting interpretation of the body of evidence presented.

1.1 Response: We thank the reviewer for their concise summary of our work. We are pleased that the reviewer found our results compelling and agreed that the California condor is an ideal study species to tackle the important questions this paper addresses.

1.2 Comment: The demonstration of a positive effect of outreach also is an important result. Previous work has shown that outreach increases awareness of conservation issues and needed management actions, but this increased awareness generally cannot be linked to conservation success or changes in human behavior. The results of this study make both of these links. Nevertheless, I think the authors sometimes highlight this conclusion at the expense of their conclusion about the effect of the bans on lead ammunition. The latter action is the more controversial one, hence evidence of its effectiveness is critical and should be highlighted more in my opinion.

1.2 Response: We appreciate the reviewer’s opinion on the emphasis of our manuscript. We have made edits to emphasize the importance of our finding that bans on lead ammunition have been effective at reducing lead exposure, and thus lead-related mortality, of California condors in California. In particular:

In “Broader Significance”, we now focus exclusively on lead ban effects for the first half of the paragraph (lines 293-302):

California is currently the only U.S. state to ban lead-based ammunition for taking wildlife, and substantial resources are expended on outreach to encourage the use of nonlead ammunition. The success of these conservation actions can serve as a bellwether for addressing lead poisoning ~~in~~ of wildlife elsewhere, in particular for the United Kingdom⁶⁴ and European Union^{65,66}, ~~which where~~ are both currently considering legislative restrictions on the use of lead ammunition have recently passed or are under consideration. More generally, our findings can inform approaches to addressing the global poisoning crisis for ~~of~~ vultures and other scavengers poisoning globally^{42,43,67,68}.

~~W~~Overall, we show that despite increases in condor lead exposure and mortality, lead ammunition bans and outreach in California have been highly effective, and remain so, but countervailing forces have masked this efficacy.

We then have two sentences on outreach, before closing with sentences that highlight both legislation and outreach (lines 311-313):

We were able to leverage detailed, long-term data sets to show that for condors, a classic slow life history species, legislation and outreach played a clear, measurable role in reducing lead exposure and thus increasing survival.

We also have now further emphasized ban effects in our edits to the legend for Fig. 3.

1.3 Comment: Clarity of presentation is generally good, but I found the information provided about pig hunting sufficiently lacking to seriously compromise my ability to interpret results. What is lacking is information about use of lead ammunition in pig hunting. Is the use of lead ammunition legal for pig hunting? If not, what can account for pig hunting leading to more exposure to lead than deer hunting? What is the difference between pig hunting and feral pig depredation? If depredation refers to killing of pigs by non-human predators, why would that result in exposure to lead?

1.3 Response: Wild pigs in California are hunted for meat and hunters are required to purchase hunting tags. However, wild pigs are also considered a pest species – causing substantial damage to crops – and are thus killed by land owners and managers under separate rules than those that govern sport hunting. As we now clarify in the text (see below), records on these culling activities have not been consistently collected over long time spans, preventing their use in our models. In addition, we recognize that although ‘depredation’ is the term used in the U.S. for lethal control of pest species, it can also be used to refer to nonhuman predation. Thus, to avoid confusion, we have exchanged our use of ‘depredation’ for ‘culling’ and ‘control’ throughout our manuscript. We have also used the term ‘wild pigs’ to clarify that we are not referring to livestock.

We have added the following information related to the use of lead ammunition for pig hunting in the main text and methods.

Main text (lines 164-166):

Hence, the use of lead ammunition for deer and wild pig hunting as well as control of wild pigs or ground squirrels became illegal in condor range after 2008 and throughout California after 2019.

Main text (lines 183-185):

However, available evidence suggests ~~feral-wild~~ pig populations are growing and expanding their range in California^{54,55}, likely leading to greater control efforts (Supplementary Fig.- 4).

Main text (lines 199-205):

Non-target mortality from culling is a global concern^{56,57,58}, and the use of lead ammunition for culling wild pigs, although illegal in California, is likely no exception. We lacked data to evaluate direct links between condor lead risk and pig culling, but we hypothesize that it is a major driver in recent increases given the apparent ongoing range expansion of wild pigs in California⁵⁵ coupled with our findings of increased condor lead risk associated with pig hunting, correlations between pig hunt and pig cull levels, and increasing trends in the ratio of pig cull to pig hunt levels for Bioyears 20167 - 2022 (Supplementary Fig. 4).

Methods (lines 466-474):

Culling of wild pigs occurs mainly on private land, where pigs can cause widespread property damage. Permitting and reporting regulations have changed over the past 20 years such that reliable records of cull levels are unavailable before 20176 (Supplementary Fig 2 g--h) preventing analysis of pig culling in condor lead risk models. However, we calculated correlations between pig hunting and culling levels (corr function in R⁷⁶) and performed simple linear regressions to assess trends in the relationship between PigCull/PigHunt (lm function in R) for Bioyears 2016 – 2022 (Supplementary Fig. 443). Bioyear 2016 only included data from Jan 2017 – Aug 2017 (vs Sep 2016 – Aug 2017) for these analyses due to lack of robust culling data for 2016 and earlier.

Finally, we added the variable *PigCull* to the variable definition table, Supp. Table 1 and added a figure and supporting analyses to explore the relationship between wild pig hunting and culling (Supp. Fig. 4).

1.4 Comment: My other criticism of the presentation is that many of the figures are very complex, requiring the reader (at least this reader) to study them for a while in order to understand them. Figure 3 is particularly difficult to understand, and the description of the relevant modeling in the methods is not detailed enough to be helpful in understanding this figure. What is a “residual trend”? (I also think there is a mistake in the legend, a (c) that should be and (a).

1.4 Response: We thank the reviewer for catching the error in labeling in Figure 3. While we recognize that our figures pack in a lot of information, we feel that simplification by removing some of the elements would hamper our ability to fully explain our results and make the manuscript less compelling. To aid in the interpretation of the figures, we have rewritten the Fig. 3 legend to be clearer and better direct the reader to the important assumptions and conclusions as follows:

Fig. 3. Drivers of blood lead exposure ~~in-for~~ California condors in California. ~~Observed condor lead exposure (‘None’, red bars) increased over time in both (a) Central and (b) Southern, but increases in (c) pig hunting by humans in Central and (b) ‘wild’ behaviors by condors in Southern drove most of the increases and masked the role of 2008 and 2019 lead-based ammunition bans in reducing exposure. Plots show year effects (categorical coefficients) and 90% confidence intervals for (a) Central and (b)~~

Southern for successively more complex linear mixed models that predict condor blood lead levels ($\ln Pb$, $n = 1421$ samples for 188 *Central* condors, 1758 samples for 179 *Southern* condors) for *Bioyears* (Sep – Aug) from 2008 onward, relative to ~~the all samples~~ years before *Bioyear* 2008. Red bars depict the ~~b~~Base model, which (‘None’, red bars) ~~accounts~~ accounts only for whether a ~~for~~ blood sample ~~was~~ targeted for suspected lead exposure (i.e., $\ln Pb \sim \text{Int} + \text{Targeted} + \text{BioYear} + 1|ID$) and thus shows the observed increasing temporal trend in lead exposure for both flocks. Remaining sets of bars show the remaining temporal trends in exposure after accounting for the added effects of other variables, thus showing how these variables account for overall changes over time. Orange bars show year effects for a model that also accounts for age and release time effects. Yellow bars show year effects for a model that adds effects of condor behavior (e.g., rates of proffered feeding) and thus illustrates annual differences not accounted for by age, release time and the increases in ‘wild’ behavior of condors. Note that increases in wild behavior drove most of the exposure increases for (b) *Southern*, as indicated by the very weak temporal trends in exposure after accounting for these effects (b). Finally, blue bars show results for models that add hunt effects and thus show how lead exposure has changed from year to year after accounting for all supported factors linked to lead exposure (i.e., targeted, age and release time, behavior, hunting); while bars for models including age and time since release, behavior, and hunting show residual trends after accounting for these explanatory effects. Pig hunting increased lead exposure in both flocks (a,b) and drove most of the increases in (a) *Central* (a). Plotting year effects for increasingly complex models unmask the strong role that the 2008 and 2019 lead-based ammunition bans, depicted by dashed vertical lines, played in lowering exposures for both flocks, depicted by dashed vertical lines. but increases in (c) pig hunting by humans in *Central* and (b) ‘wild’ behaviors by condors in *Southern* drove most of the increases and masked the role of 2008 and 2019 lead-based ammunition bans in reducing exposure.

We have also adding the following text to the Methods revised the methods (lines 539-541):

Finally, we visually depicted the relative importance of different classes of variables by plotting annual variance as categorical year effects for successively more complex models nested within the best-supported model (Fig. 3).

We also note that for full transparency on all of our analyses, readers can also consult the R code we submitted as supplementary material.

1.5 Comment: My other criticisms of the paper concern the modeling analyses. In the modeling of blood lead levels (and survival to a lesser extent), a huge number of models have very similar AIC values. The authors need to provide more details in the methods about how they handled this in the context of uninformative variables, model averaging and coefficient estimates. The modeling generally is firmly grounded in data and evidence.

1.5 Response: We appreciate the reviewer's conclusion that our models are generally firmly grounded in data and evidence and their request for more information on how model selection was done in the face of very similar AICc values for multiple top models. One approach to dealing with similar AICc scores is to do model averaging. However, we chose not to model average our parameter estimates, because of the presence of numerous interactions in some models, which make the biological interpretation of model averages extremely difficult in this situation. Nonetheless, we share the reviewer's concerns about appropriately conveying output of our analyses that include large numbers of models. We are confident in our results because we are primarily using them to identify the drivers of lead exposure. Also, by providing coefficient estimates in our model selection tables, we show that coefficient estimates are quite consistent across models, especially for the drivers that we are most interested in dissecting with our work (e.g., hunting, lead bans). However, we have made several revisions to address the reviewer's concern:

- Simplified AICc tables for lead exposure models to only include the top model set ($\Delta AICc \leq 2$)
- Shaded models that indicate the likely presence of uninformative parameters in model selection tables. We explain this shading in the methods and table legend.
- Calculated three metrics of variable importance, which we plot in a new supplemental figure (Supplementary Fig. 3). We provide the following new methods for this figure in Methods, including condor *ID*, a repeated measure.

Methods (lines 524-543, lines 539-541 is also in response to prior comment)

We used model AICc to guide model selection, comparing results for all possible combinations of variables and interactions selected *a priori* via the dredge function in the R package MuMIn⁷⁸. We also calculated and plotted (Supplementary Fig. 3) several measures of variable importance to guide our interpretation of the drivers of lead exposure. Specifically, we calculated the sum of AICc weights for all models that contained each predictor variable (MuMIn, sw function), a widely used method, but one that has recently been both criticized as imprecise and also defended^{79,80} but see⁸¹. We also ran models using standardized coefficients (centered and multiplied by the partial standard deviation, which adjusts for multicollinearity among variables, using the `beta = "partial.sd"` option in the dredge function) and calculated model-averaged coefficients for these standardized variables (model.avg function). We used these coefficients to calculate variable importance as the ratio of the absolute value of the model-averaged coefficient to the maximum model-averaged coefficient, as recommended by Galipaud et al.⁷⁹ and Cade⁸⁰, using AICc weights for all models ("full") and only the subset of models containing the variable ("subset"). Finally, in our model selection tables, we indicate models that have higher ranked models nested within them and thus contain parameters

that are likely to be uninformative^{82,83} (MuMIn, nested function). Finally, we visually depicted the relative importance of different classes of variables by plotting annual variance as categorical year effects for successively more complex models nested within the best-supported model (Fig. 3). We included condor ID, a repeated measure, -and Bioyear as crossed random effects in all models, except when depicting annual changes (i.e., Fig. 3, omits Bioyear as a random effect).

We added the following to the main text when we reference the variable importance figure:

To better understand the paradox of rising condor lead exposures in California despite significant efforts to limit lead-based ammunition use, we fit linear mixed effects models (LMMs) and calculated variable importance metrics (Supplementary Fig. 33) to dissect the different and sometimes opposing influences (e.g., condor behavior, hunter behavior, post-ban time periods, outreach effort) on condor blood lead levels (Supplementary Tables 3, 5, 9-12).

Analyses of these LMM model sets show that while multiple models had low AICc values, explanatory variables in the best models were generally the ones most supported across all models and the same patterns of effects were stable across the well-supported models (Supplementary Figure 34, Supplementary Tables 9 – 12).

1.6 Comment: The one exception is the modeling of contaminated meal exposure. There are a number of unsupported assumptions in this analysis such that I found the validity of the results questionable. This analysis seems to me somewhat of a tangent in that it attempts to provide a level of detail about lead exposure that is not necessary to the major conclusions of the study. Because the results essentially duplicate the blood lead level results, I found them uninformative. I suggest deleting this analysis.

1.6 Response: While we agree with the reviewer that the two statistical analyses are very similar in their use to assess the importance of different drivers of lead exposure in condors and thus may appear duplicative, we chose to employ both analyses as they do give different results that are important for our work. We used glmer to identify the predictors of blood lead exposure because this is a standard method with straightforward model selection theory that should have the greatest power to screen large model sets. In contrast, we developed the Poisson-Gamma modeling approach to estimate the scale of the meal contamination problem (the rate of encounter of lead-contaminated meals, separated from the amount of lead received per meal), as this is critical information for management that can't be derived from the simpler glmer analyses. The results from these models confirm earlier, but not as well-derived, estimates that predicted that very few contaminated meals are enough to result in ongoing poisonings and declining condor populations (Finkelstein et al., 2012).

Because of the importance of the contaminated meal encounter rate estimates, we have elected to leave the Poisson-Gamma models in the manuscript, but we have attempted to address the reviewer's concerns in several ways:

First, we have minimized discussion of the duplicative results from these analyses in the main text and only discuss the results that are novel.

Second, we have added a caveat to the main text that these models are more complex and require several statistical assumptions that are not possible to test, and we have expanded the methods which discusses these assumptions and their possible importance for our results.

Main text (lines 242-244):

While these models rely on multiple statistical assumptions (see Methods: Models to predict rates of contaminated meal exposure), they provide a way to directly estimate encounter rates, a key factor that management interventions seek to influence.

Methods (lines 585-592)

Poisson-Gamma models are predicated on several statistical assumptions that are difficult to directly test with our data. Most notably among these is that encounters are independent and well-described by the Poisson distribution, and that individual encounters contribute to the summed observed blood lead levels seen in a sample with Gamma distributed values. Both these assumptions are reasonable, but without exceptionally detailed data on individual contaminated meals, they are impossible to test. One process we could not model separately and that must be acknowledged is ~~was~~ the attenuation of lead present in the blood following feeding on a contaminated carcass.

Methods (lines 598-599)

We assumed that predictors influenced the mean of the Poisson and Gamma distributions, but that ~~We estimated~~ the shape parameter of the Gamma distribution as ~~is~~ a constant.

We have also added main text to underscore the broader importance of these findings.

Main text (lines 262-267)

We have previously shown that current rates of lead on the landscape are enough to transition condors in California from self-sustaining to conservation reliant⁴⁵. Overall, our finding that this mortality level results from fewer than ten contaminated meals per year helps explain why lead poisoning of scavengers remains a global conservation problem^{61,62,63}.

REVIEWER #2 (REMARKS TO THE AUTHOR):

2.0 Comment: This manuscript makes uses of extensive datasets to explore why recovery efforts for the California Condor in the U.S. appear to be stalled due to increases in lead poisoning, despite a ban on lead ammunition in California and extensive outreach/education efforts about the use of non-lead ammunition. The authors show that an increase in more natural behaviors by condors (i.e., feeding from non-subsidized food, and moving over large distances) and an increase in hunting pressure together mask the success of lead ammunition bans and outreach campaigns.

The authors brought a myriad of data to address their study objectives, and I appreciated the depth they went to look at this issue. I think the authors could strengthen the MS by considering changes to the framing of the study and broadening the implications of their study beyond the California Condor. I expand on these and other comments below.

2.0 Response: We have added text and 23 new citations throughout our manuscript to frame our study in a more general context. We describe specific examples below.

2.1 Comment: The idea of describing the study as one that is illustrative of Red Queen dynamics is interesting, but given Nature Communications is a multi-disciplinary journal I suspect it will not resonate with a broader audience. What's more, the authors do a poor job of explaining what is meant by Red Queen dynamics by largely focusing on a quote from Alice in Wonderland, which is bound to be confusing to some. Therefore, I suggest dropping the Red Queen piece and instead focus on the idea that species recovery can be masked by temporal changes in behavior, which I expect to resonate with readers much more than Red Queen dynamics.

2.1 Response: We acknowledge that in our original submission we didn't explain the idea of the Red Queen well enough for some readers to understand. However, this term has found wide application in a range of scientific disciplines at this point. The original use of the term Red Queen dynamics in fields close to conservation ecology was in the field of evolutionary biology. Riffing off a line from Alice in Wonderland, it describes a process whereby intense competition drives innovations that lead to speciation, but little change in fitness. However, it is now widely recognized and used across academic disciplines to describe adaptation to ongoing system changes and thus should be broadly resonant in a multidisciplinary journal. To cite just a few examples:

- Health claims in food marketing: Cuevas, S., N. Patel, C. Thompson, M. Petticrew, S. Cummins, R. Smith, and L. Cornelsen. 2021. Escaping the Red Queen: Health as a corporate food marketing strategy. *SSM - Population Health* 16:100953.
- Doping in sports: Uyar, Y., A. Gentile, H. Uyar, Ö. Erdeveciler, H. Sunay, V. Mîndrescu, D. Mujkic, and A. Bianco. 2022. Competition, gender equality, and doping in sports in the Red Queen effect perspective. *Sustainability* 14:2490.
- Expectations on minority faculty: Negatu, S., M. Arreguin, K. Jurado, and C. Vazquez. 2022. Being the Alice of academia: lessons from the Red Queen hypothesis. *Pathogens and Disease* 80:ftac034.
- Coercive bargaining dynamics: Amadae, S. M., and C. J. Watts. 2023. Red Queen and Red King Effects in cultural agent-based modeling: Hawk Dove Binary and Systemic Discrimination. *The Journal of Mathematical Sociology* 47:283-310.
- Competition among multinational corporations and banks:
 - Chiao, Y.-C., C.-C. Lin, and Y.-C. Chang. 2024. Run, not walk: advanced red queen effect and mutual forbearance effect in multimarket contact. *International Journal of Emerging Markets*.
 - Lee, S., Y. Kwon, N. N. Quoc, C. Danon, M. Mehler, K. Elm, R. Bauret, and S. Choi. 2021. Red Queen Effect in German Bank Industry: Implication of Banking

Digitalization for Open Innovation Dynamics. *Journal of Open Innovation: Technology, Market, and Complexity* 7:90.

- Provisioning of public goods: Salahshour, M. 2021. Evolution of cooperation in costly institutions exhibits Red Queen and Black Queen dynamics in heterogeneous public goods. *Communications Biology* 4:1340.
- Retail sector innovation: Cuthbertson, R., O. A. Rusanen, and L. Paavola. 2023. *The Red Queen Retail Race: An Innovation Pandemic in the Era of Digitization*. Oxford University Press.

The concept is also used in popular culture. For example, in the song *Legacy*, Jay-Z references it to describe the struggle for black liberation: “That’s called the Red Queen’s Race/You run this hard just to stay in place/Keep up the pace, baby/Keep up the pace.”

Given the pace of global change, we feel this term is a useful addition to the conservation lexicon to describe dynamics in which changes in the threat landscape require adaptation by conservation managers. We also note that since the other two reviewers did not object to this term, we suspect that it does have broader resonance. However, to address the reviewer’s call for clarification, we have added the following edits to the introduction to provide context:

Main text (lines 38-49)

Recovery actions also ~~occur~~ take place in nonstationary ecological settings, in which behavioral shifts by recovering or co-occurring species, as well as by humans, can increase threat levels, masking the success of even highly effective measures. When worsening threats mask conservation efficacy, this sets up a classic Red Queen dynamic. The term Red Queen dynamics, which references Lewis Carroll’s *Through the Looking Glass* (“Now, here, you see, it takes all the running you can do, to keep in the same place”^{15,16}), was originally employed in the biological literature to describe coevolutionary processes¹⁷ but today is widely used to describe situations in which individuals or organizations must adapt to keep pace with ongoing system changes^{18,19,20}. Given the speed of global change, Red Queen dynamics are likely to play an increasingly prominent role in species conservation, and correctly understanding these dynamics will be masking the success of even highly effective measures. This masking is will be particularly ~~problematic~~ important when conservation actions are costly or controversial and will likely be increasingly common for species conservation given the speed of global change.

Comment 2.2: The MS has an extensive analysis of condor data, but in places it is rather limited beyond this species. Although the Introduction provides information that goes beyond condors, the Discussion does not, and instead is almost exclusively focused on condors. Towards the end of the MS the authors try to argue that their findings for condors likely apply to other systems, but they also note that the data available for this species is exceptional and that few other species have such a wealth of data available. That makes me question whether the impact of this study is restricted mostly to condors, or if it goes further and can inform us about other systems. For

broad readership in a multidisciplinary journal it has to be the latter, and therefore I suggest the authors should make the Discussion more robust in this regard.

Response 2.2: We agree that we did not do enough to highlight the broader implications of this work, and have made the following revisions throughout to showcase how our results can inform other species and systems:

Introduction

(Main text lines 51-52) - *Added references from reintroduction literature:*

For reintroduced populations, understanding how changing behavior influences population dynamics can be critical^{21,22,23}

(Main text lines 56-58) - *Added connection to rewilding efforts:*

Evaluating interactions between wild behavior and threat exposure is increasingly important as ~~the field of~~ conservation seeks to maximize the value of human-modified landscapes^{29,30} and some voices call for global rewilding^{31,32,33}.

Results and Discussion (also in response to Reviewer 1)

(Main text line 199) - *Added connection to non-target mortality from culling*

Non-target mortality from culling is a global concern^{56,57,58}

(Main text lines 262-267) - *Added link between meal contamination rate estimates and global lead poisoning*

We have previously shown that current rates of lead on the landscape are enough to transition condors in California from self-sustaining to conservation reliant⁴⁵. Overall, our finding that this mortality level results from fewer than ten contaminated meals per year helps explain why lead poisoning of scavengers remains a global conservation problem^{61,62,63}.

Broader Significance:

(Main text lines 312-318) - *Expanded our closing paragraph*

Even though many species face similar threats, ~~Although few species have the same wealth of data~~ are as well-studied as condors, and this wealth of data supported unique insights on effective policy to address lead poisoning of wildlife, a global conservation issue^{56,61,62,69,70,71,72}. More broadly, our findings ~~serve as an example and~~ underscore the importance of comprehensive analyses when evaluating the success of recovery -actions in complex settings, especially those with changing behaviors and threats that may result in Red Queen dynamics.

Comment 2.3: The authors state that one of the key findings from the study is that outreach activities have been highly effective at reducing lead exposure from hunting, but the data that are used are observational (i.e., the number of people contacted at outreach events, and the number of non-lead ammunition distributed) and require untestable assumptions. For example, increases in the distribution of non-lead ammunition does not necessarily mean that such ammunition was used for hunting the focal species (deer or pigs) that often contaminate condors with lead. Therefore, my view is that more caution is needed when drawing conclusions about the effect of

outreach activities. Additionally, because there are ostensibly no detailed data on human behaviors related to these outreach efforts (e.g., how often distributed ammo was used for hunting pigs/deer by those who received it), the authors would do well to highlight the need for new studies focused on this topic to bring a more comprehensive understanding of how outreach activities impact condor recovery.

Response 2.3: We agree that new studies are needed to bring a more comprehensive understanding of this topic, and especially to make even better links between outreach, how they influence the details of human behavior, and the resulting effects on conservation targets. However, we also feel that it is difficult to imagine a set of spurious effects operating in our system that would be making the general conclusions we draw about legislative and outreach effects incorrect. In addition to adding a call for more studies of these effects, we have also added details on the outreach data in the supplement and methods, as follow:

Supplementary Discussion

Although we assume—and our models indicate—that higher outreach is associated with higher use of nonlead ammunition, future studies are needed to validate this assumption. In addition, since our outreach data started after the first ban (2008), we only analyzed the effects of outreach occurring after the implementation of lead bans. Therefore, our results assume the presence of a legislative ban and do not necessarily apply to the effectiveness of outreach when compliance is voluntary.

Methods (lines 488-491)

Although we do not have data that directly measures the amount of nonlead ammunition used per contact or box of ammunition distributed, we make what we believe is the reasonable assumption that non-lead ammunition use is positively related to numbers of contacts and numbers of boxes of ammunition distributed.

Comment 2.4: The authors used the Baja population as a control for the U.S. populations, but it was noteworthy that this population has $\lambda < 1$ in Fig. 2b despite lead ammunition not being a concern. It might seem a bit beyond the scope of this MS, but I think readers will be curious about why that's the case, and whether the factor(s) in Baja that prevent recovery may also be present in the U.S. population and have the potential to interact with lead poisoning. This is important because even if lead poisoning is reduced through recovery efforts in the U.S., addressing the other factor(s) are likely to be as important – or more important – for effective long-term recovery of the species.

Response 2.4: We respectfully correct the reviewer's misreading of our Figure 2. Figure 2 is depicting survivorship, not growth rate. The growth rate for the Baja population is 1.017 as stated in our text (lines 107-110):

In contrast, *Baja* has had very low blood lead levels (Fig. 2a), has experienced essentially no lead-related mortality (Fig. 2c), and has an increasing population ($\lambda = 1.017$ if excluding contributions from captive-bred releases, Supplementary Tables 2, 8).

REVIEWER #3 (REMARKS TO THE AUTHOR):

Thanks for the opportunity to review this manuscript. Overall, I think the RQs are well articulated, the methods are sound, and the findings are interesting and (mostly) well contextualized in the literature. I have only two substantive concerns:

Comment 3.1: The increasing "wilding" behavior of breeding program condors has been studied in other contexts and you might do well to engage with new literature on the species, even look to the social science and humanities literature on condors. I might suggest Farnsworth, J. S. 2015. The Condor Question Revisited as one example. I think you could contextualize and situate your findings a bit more in the rich world of condor research.

Response 3.1: We have added several references related to rewilding (also in response to reviewers 1 and 2 to underscore how our work can inform other species and study systems) in the introduction, as follows:

Main text (lines 51-58)

For reintroduced populations, understanding how changing behavior influences population dynamics can be critical^{21,22,23}. An often-implicit assumption of endangered species recovery is that restoring 'wild'²⁴ behavioral traits ~~following for released individualseaptives release~~ will increase their fitness^{25,26,27}. While expression of wild behavior is often desirable for species recovery, it can entail moving out of protected areas or utilizing more dangerous resources²⁸. Evaluating interactions between wild behavior and threat exposure is increasingly important as ~~the field of~~ conservation seeks to maximize the value of human-modified landscapes^{29,30} and some voices call for global rewilding^{31,32,33}.

We reviewed Farnsworth 2015 but are uncertain how this work is related to our study. Citing Farnsworth 2015 would require the introduction of new concepts and substantial text to incorporate social sciences and humanities literature. Given the need to restructure our manuscript to do this, we respectively prefer to not make this major shift in the emphasis of our paper.

Comment 3.2: I suggest you craft a paragraph addressing all threats to your internal validity (of your data metrics and operational definition). You could say a bit more, too, regarding generalizability but the first part is more important.

Response 3.2: Although we are unclear specifically what the reviewer's concerns are with respect to our internal validity and generalizability (external validity), we feel it is more comprehensible to discuss caveats and assumptions as they come up in their presentations of the data streams, especially since we have multiple types of data and analyses. However, we will defer to the editor if preference is for a stand-alone paragraph. In addition to existing text describing assumptions and caveats, we have added text to more clearly describe the threats to our internal validity to the main text, methods, and extended data.

Internal validity

- We note that we have a uniquely complete set of monitoring data for all individual condors in California for their entire lifetimes in the wild. This near-complete dataset minimizes issues related to selection bias and many other factors that limit internal validity.
- We have used mixed models to account for inter-individual variation
- As we describe, we have also attempted to obtain all available data on hypothesized drivers of lead exposure and mortality and to investigate plausible interactions. Through this approach we have made every effort to identify and account for confounding factors and spurious correlations.
- To expand on the relative importance of different drivers of lead exposure, we have added metrics of variable importance and highlighted models with uninformative parameters.
- We have added to text to clarify our assumptions regarding correlations between outreach levels and use nonlead ammunition. We note that future study on these linkages is needed. We also note that we do have data on some factors that may influence condor lead exposure such as ground squirrel control and wild pig culling.
- We have added text and analyses to support our assumption that pig culling is positively correlated with pig hunting and likely increasing.

Generalizability.

We have added text to highlight the broader significance of our findings to conservation, including relevance to non-target mortalities associated with culling programs, lead poisoning effects on scavengers globally, and more generally, to the need for careful analyses due to the likely increase of Red Queen conservation dynamics.